# The usherin mutation c.2299delG leads to its mislocalization and disrupts interactions with whirlin and VLGR1

Lars Tebbe[1,4], Maggie L. Mwoyosvi [1,3,4], Ryan Crane[1], Mustafa S. Makia[1], Mashal Kakakhel [1], Dominic Cosgrove[2], Muayyad R. Al-Ubaidi [1] ✉ & Muna I. Naash [1] ✉

Usher syndrome (USH) is the leading cause of combined deafness-blindness with type 2 A (USH2A) being the most common form. Knockout models of USH proteins, like the *Ush2a*[-/-] model that develops a late-onset retinal phenotype, failed to mimic the retinal phenotype observed in patients. Since patient's mutations result in the expression of a mutant protein and to determine the mechanism of USH2A, we generated and evaluated an usherin (USH2A) knock-in mouse expressing the common human disease-mutation, c.2299delG. This mouse exhibits retinal degeneration and expresses a truncated, glycosylated protein which is mislocalized to the photoreceptor inner segment. The degeneration is associated with a decline in retinal function, structural abnormalities in connecting cilium and outer segment and mislocaliztion of the usherin interactors very long G-protein receptor 1 and whirlin. The onset of symptoms is significantly earlier compared to *Ush2a*[-/-], proving expression of mutated protein is required to recapitulate the patients' retinal phenotype.

Usher syndrome (USH) is a recessively inherited form of combined loss of vision and auditory function, often accompanied by vestibular dysfunction[1,2]. Based on the onset and severity of the symptoms, USH is clinically divided into three subtypes USH1, 2, and 3, with USH1 being the most severe and USH2 the most common subtype[3,4]. Three genes are connected to USH2 (*USH2A*, *USH2C*, and *USH2D*), of which mutations in *USH2A* are most prevalent[3]. These *USH2A* mutations are associated with congenital moderate-to-severe hearing loss, pre-to-post pubescent onset retinitis pigmentosa (RP), and unaffected vestibular function[5]. To date, there are over 1500 different *USH2A* mutations with more than 690 variations identified, with c.2299delG being the most common one[5,6]. The deletion of the guanine at position 2299 results in a frameshift yielding a truncated protein with additional 20 amino acids that are not part of the *USH2A* open reading frame followed by a translation stop codon[7–9].

Expression of the *USH2A* gene results in two isoforms of the usherin protein: the short isoform A corresponds to a 170 kDa secreted protein coded for by 21 exons, and the long isoform B is a 580 kDa transmembrane protein coded for by 72 exons[10,11]. Although the precise function of usherin is still under investigation, there is evidence to suggest that it binds to other USH proteins generating a large interactome complex (USH interactome) at the connecting cilium (CC)/periciliary region of photoreceptors and at the ankle links of hair cell stereocilia[12,13]. This complex plays an important role in the proper development and structural maintenance of these cells[12,14–16]. The interaction between usherin, the very large G-protein coupled receptor 1 (VLGR1, *USH2C*, or ADGRV1), and the PDZ scaffold protein whirlin (WHRN, *USH2D*) was found to be essential for the stability of the complex[12,14,16]. In the retina, usherin interacts with VLGR1, WHRN, and SANS to form the periciliary membrane complex (PMC) essential for the loading of cargo at the photoreceptor cilium[12,17,18]. Knockout of either USH2A (*Ush2a*[-/-]) or VLGR1 (*Ush2c*[-/-]) led to the depletion of the other from the interactome in the periciliary region of the photoreceptor[12,16]. Support for the idea that the USH protein network

[1]Department of Biomedical Engineering, University of Houston, Houston, TX 77204, USA. [2]Boys Town National Research Hospital, Omaha, NE, USA. [3]Present address: Department of Microbiology & Immunology, University of Oklahoma Health Sciences Center, Oklahoma City, OK 73104, USA. [4]These authors contributed equally: Lars Tebbe, Maggie L. Mwoyosvi. ✉e-mail: malubaid@central.uh.edu; mnaash@central.uh.edu

plays a role in cargo transport in the retina comes from the observation that transducin translocation is delayed in the *shaker 1* mouse (a model for *USH1B*) and in the *whirler* mouse (a model for *USH2D*)[17,19,20]. While a genotype/phenotype correlation exists for USH2A, removing or altering any piece of this complex will ultimately lead to RP-associated structural and/or functional defects[2,12,14,16,21–23].

The model of choice for studying USH2A has been the usherin knock-out mouse[21]. Although this model is useful for mimicking the hearing loss detected in patients and provided some insight into the function of usherin in the retina, the onset of the retinal phenotype observed in this model is much later than that seen in patients[21]. Thus, we generated a knock-in (KI) mouse line, which carries the common human c.2299delG frameshift mutation (corresponding to c.2290delG in mouse) to further our understanding of the pathogenesis of *USH2A*-associated c.2299delG in the retina.

We found that this model exhibits vision loss associated with retinal degeneration and morphologic changes to the photoreceptor cilium, consistent with the late-onset retinitis pigmentosa phenotype observed in USH2A patients[5,6]. Furthermore, we show that truncated usherin is expressed (~110 kDa), glycosylated and intracellularly mislocalized to the inner segment, which led to mislocalized VLGR1 and WHRN. Since we found it to be glycosylated but largely excluded from the plasma membrane and extracellular fractions of the *Ush2a^{delG/delG}* retina, we hypothesize that it is likely trapped in intracellular vesicular structures. The absence of usherin, VLGR1, and WHRN at the connecting cilium in the KI retina provides evidence for the disruption of the USH2 complex at the periciliary region that leads to mislocalization of opsins and reduction in outer segment (OS) width.

## Results

### C.2290delG results in a truncated intracellularly localized protein

The strategy presented in Supplementary Fig. 1A was used to generate the KI mouse line carrying the c.2290delG mutation. The c.2290delG KI mouse incorporates the deletion of a guanine at position c.2290, which is equivalent to position c.2299 in humans. Similar to patients, this causes a frameshift that results in an amino acid change from glutamine to serine at position 764 in mouse (767 in humans). In humans, this frameshift is followed by the addition of 20 amino acids before a stop codon (Supplementary Fig. 1A, B). To accurately model the human genetic situation in the mouse, we also inserted the sequence coding for the additional human 20 amino acids extension immediately after the c.2290 deletion followed by a triple (3x) FLAG-tag to facilitate differentiation between mutant and wild type (WT) usherin. To help understand the cellular expression of *Ush2a*, the stop codon was followed by an internal ribosomal entry site (IRES) and GFP coding sequence (Supplementary Fig. 1A). The IRES allows for the transcription of a bicistronic message, and the translation of two independent proteins, the truncated usherin and GFP (Supplementary Fig. 1D). Although the phenotype associated with the c.2299delG mutation in humans is autosomal recessive, the retinal phenotype in mice both homozygous and heterozygous for the c.2290delG KI allele (herein referred to as *Ush2a^{delG/delG}* and *Ush2a^{delG/+}*) were evaluated in this study.

Following the generation of the *Ush2a^{delG/delG}* mouse model, we performed qRT-PCR on retinas using primers that recognize both the mutant and WT transcripts. While there was a significant reduction in usherin transcript at postnatal (P) 15 (23.5% reduction in *Ush2a^{delG/delG}*, **$p < 0.01$), we found that total *Ush2a* transcript levels at P1, P30 and P360 in the *Ush2a^{delG/delG}* are equivalent to those in WT. This suggests that the KI gene is transcribed at levels equivalent to that in WT throughout the life of the animal (Fig. 1A). In both WT and KI mice, there is an increase in retinal *Ush2a* transcripts during early development, peaking at P15, after which levels drop by half at P30 and stabilize thereafter. Using primers specific to WT *Ush2a*, we confirmed that no WT transcript was detected in *Ush2a^{delG/delG}* retinas and that no

KI transcript was detected in P30 WT retinas using primers specific for the KI allele (Supplementary Fig. 1C).

To characterize the mutant protein, we performed immunoblot (IB) on retinal extracts from P30 *Ush2a^{delG/delG}* and WT mice and showed that the KI transcript resulted in a truncated USH2A protein ~110 kDa in size as detected by an anti-FLAG antibody (Fig. 1B). The analysis also confirmed the expression of GFP in the *Ush2a^{delG/delG}* model (Fig. 1B). In the WT retina, neither the truncated USH2A protein nor GFP were detected on IB (Fig. 1B) or immunofluorescence (IF) (Supplementary Fig. 1E, upper images). To determine whether the mutant USH2A can form intermolecular-disulfide linked complexes or non-covalent complexes with itself or with other proteins, we performed IB under reducing and non-reducing conditions on retinal extracts from P30 *Ush2a^{delG/delG}* mice either under native conditions (without SDS, Fig. 1C) or in the presence of SDS (Fig. 1D). In all cases, the mutant USH2A protein ran at the same size of ~110 kDa, suggesting that the truncated usherin is not able to form non-covalent or covalent associations with itself or other proteins to form larger complexes. PRPH2 is shown as a control protein that forms disulfide-linked complexes detectable under non-reducing conditions (Fig. 1D, lower panel).

Previous studies showed that usherin is mainly located at the periciliary ridge and some at the synaptic terminals of the mammalian photoreceptors[15,17,24–26]. To determine the cellular localization of the mutant USH2A in comparison to the WT protein, IF was performed on WT and *Ush2a^{delG/delG}* retinal sections using antibodies against the C-terminal region of full-length usherin and against FLAG (Fig. 1E, F). In line with previous studies, full-length WT usherin was detected at the periciliary region in WT sections, partially co-localized with acetylated tubulin (acTub, marker for CC and proximal axoneme), as well as at the synapses in the outer plexiform layer (OPL, Fig. 1E, left images). High-magnification imaging of a single cilium further pinpointed the localization of WT usherin adjacent to the ciliary base of the WT retina (Fig. 1E, middle images). As expected, full-length WT usherin was not detectable in the *Ush2a^{delG/delG}* retinal sections (Fig. 1F, left and middle images). In stark contrast to WT usherin, the c.2290delG mutant USH2A (detected by an anti-FLAG antibody) was only seen in the inner segment of the *Ush2a^{delG/delG}* retina (Fig. 1F, right images) while none was observed in the WT retina (Fig. 1E, right images). Co-labeling for acTub showed the distinct absence of c.2290delG USH2A from the periciliary region (Fig. 1G), but its localization to the inner segments. This inner segment mislocalization was further confirmed by labeling *Ush2a^{delG/delG}* retinal sections with an antibody generated against the 20-amino acid human extension caused by the c.2299delG mutation (Supplementary Fig. 1E, left lower image). Co-labeling for the inner segment protein Syntaxin 3 (STX3) followed by high-resolution imaging with airyscan supported the restriction of the truncated protein to the inner segment (Fig. 1H). Additionally, 3D reconstructions from confocal stacks showed that mutant USH2A is present as puncta in spheroid-shaped structures, suggesting that the truncated protein is potentially encapsulated in vesicles (Fig. 1H, middle and right images).

The lack of a transmembrane domain in the truncated usherin (Supplementary Fig. 1D) as well as its absence from the periciliary region (Fig. 1F–H) suggests that it is unable to embed in the photoreceptor plasma membranes. To confirm this, fresh retinas from P30 *Ush2a^{delG/delG}* and WT animals were incubated in isotonic buffer (1x PBS, pH 7.3) to release all soluble interphotoreceptor matrix (S-IPM) proteins[27]. The retinas were then homogenized and the intracellular protein fraction was separated from the membrane-bound protein fractions via ultracentrifugation. Blots were probed with markers for each fraction to confirm appropriate separation (IRBP, PRPH2, and GAPDH for the S-IPM[28], membrane-bound fraction[29], and cytosolic fractions, respectively). WT usherin is restricted to the membrane-bound fraction, and is absent from either the S-IPM or the cytosolic fraction (Fig. 1I). In contrast, the mutant USH2A protein is found in the cytosolic fraction as well as in the membrane fraction (Fig. 1K),

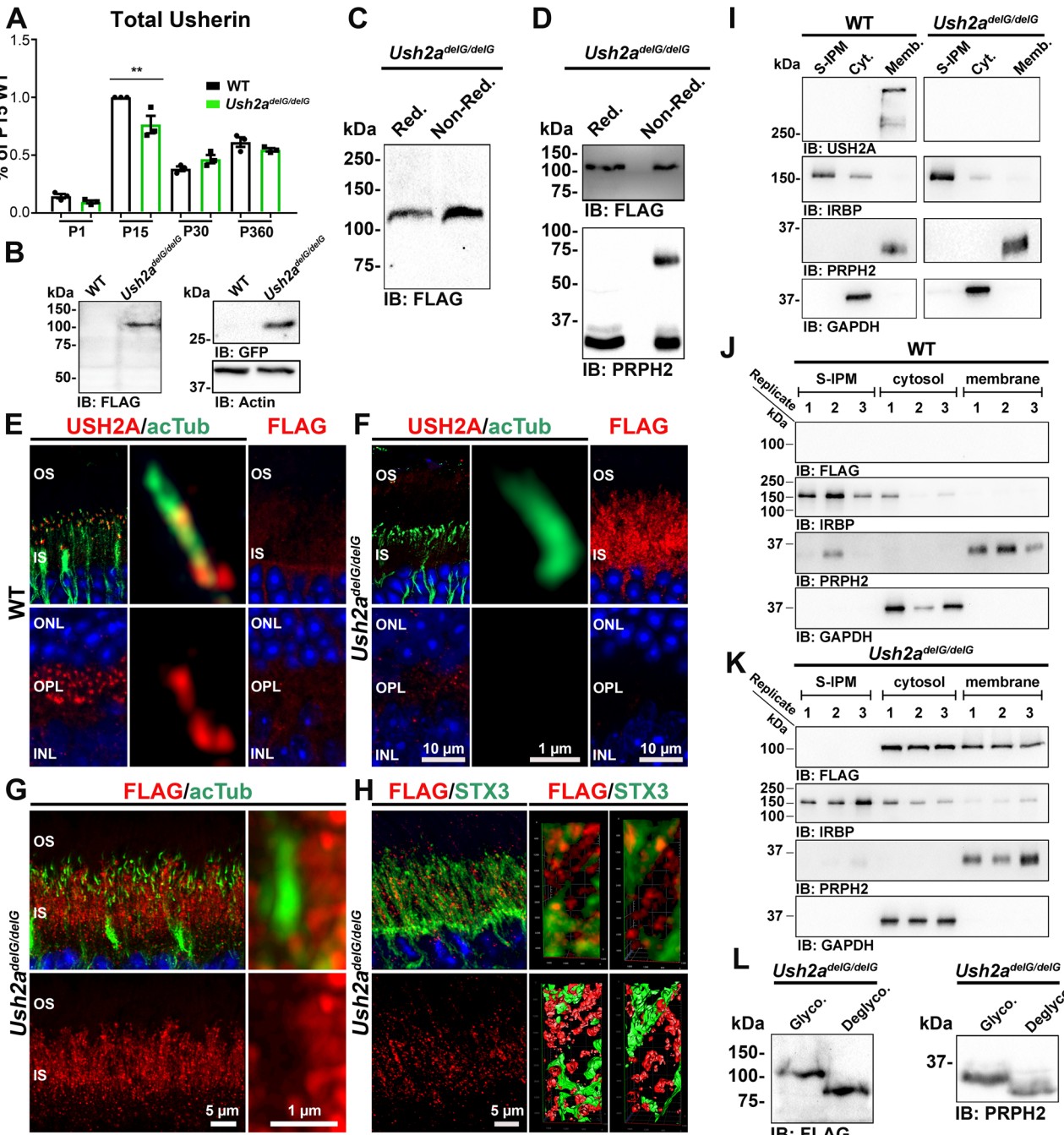

**Fig. 1 | Ush2a^delG/delG protein is stable, developmentally regulated, and mislocalized. A** Developmental steady-state levels of total usherin transcripts are comparable in WT and *Ush2a^delG/delG* retinas. Shown are mean ± SEM from three independent samples. Statistical significance was determined by one-way ANOVA. *P* values: P15: *p* = 0.0043 (**). *N* value: 3 mice per time point and genotype and measurements were made in triplicates. **B** Immunoblotting shows that the mutant protein and GFP are expressed only in the KI retinas. **C** Immunoblot shows that the mutant protein is unable to form complexes through electrostatic interactions. **D** Immunoblot under reducing and non-reducing conditions shows that truncated usherin is unable to form a covalent complex (upper panel). PRPH2 was used as a positive control (lower panel). **E** Full-length usherin is located at the ciliary base (upper left and middle images) and OPL (lower left image). The truncated protein is absent from the P30 WT retina (right images). **F** Co-staining of retinal sections from P30 *Ush2a^delG/delG* for usherin and acTub shows the absence of full-length usherin (left and middle images). Labeling for FLAG reveals the truncated protein to be localized in IS (right images). **G** Co-labeling for FLAG and acTub at P30 shows a dispersed localization of the mutant protein in the IS, without distinct localization in the periciliary region. **H** Co-labeling for FLAG and STX3 demonstrates the accumulation of the truncated protein in the IS (left images). 3D-remodeling showing truncated usherin organized in compartments (right images). **I–K** Immunoblots of fractionated WT and *Ush2a^delG/delG* retinal extracts showing full-length usherin localized in the membrane fractions in the WT and absence in *Ush2a^delG/delG* retina (**I**). Immunoblot probed with anti-FLAG antibody revealed the major portion of the mutated protein is in the cytosolic fraction while a minor portion is present in the membrane fraction (**K**). GAPDH was used as a cytosolic marker, IRBP as a S-IPM marker, while PRPH2 was used as a membrane marker. **L** Immunoblot probed with anti-FLAG antibody showing the mutant protein is glycosylated like PRPH2. OS outer segment, IS inner segment, ONL outer nuclear layer, OPL outer plexiform layer, INL inner nuclear layer, S-IPM soluble inter photoreceptor matrix, Cyt. cytoplasm, Memb. membrane, Glyco. glycosylated, Deglyco. deglycosylated, Red. reducing, Non-Red. non-reducing.

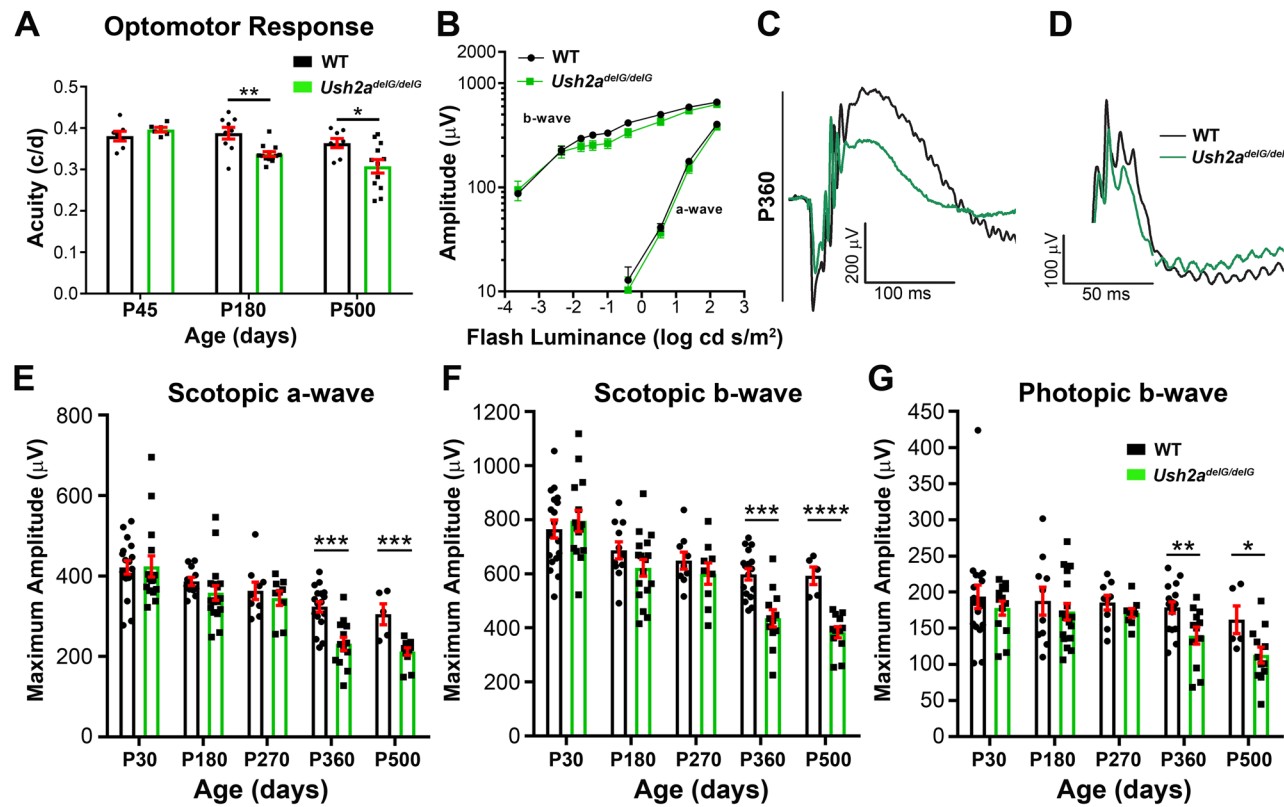

**Fig. 2 | The *Ush2a^{delG/delG}* model exhibits a late-onset decline in visual function.**
**A** Optomotor response is significantly reduced at P180 in *Ush2a^{delG/delG}* mice and persists at P500. Shown are mean ± SEM. *N* values: WT P45: 7; P180: 9; P500: 8. *Ush2a^{delG/delG}* P45: 6; P180: 13; P500: 12. Statistical significance was determined by a two-tailed unpaired *t*-test. *P* values: P180: *p* = 0.0015 (**), P500: *p* = 0.0197 (*).
**B** Maximum ERG a-wave and b-wave amplitudes plotted against the log of the eliciting flash intensities. Animals (P200) were flashed from weakest to brightest light intensities. Shown are mean ± SEM. *N* value: 7 WT mice were tested for each intensity in the measurement of both scotopic a- and b-waves. 5 *Ush2a^{delG/delG}* mice were tested for each intensity in the measurement of both scotopic a- and b-waves.
**C–G** Full-field ERGs were performed on WT and *Ush2a^{delG/delG}* mice at the indicated ages. **C, D** Representative scotopic and photopic ERG waveforms are shown from

P360 *Ush2a^{delG/delG}* and WT mice. **E–G** Maximum scotopic a-, scotopic b-, and photopic b-wave amplitudes were measured at the indicated ages. Amplitudes are significantly decreased in the P360 and P500 *Ush2a^{delG/delG}* mice. Shown are mean ± SEM. Significance was determined by a two-tailed unpaired *t*-test. *P* values: Scotopic a-wave: P360: *p* = 0.0001 (***) P500: *p* = 0.0006 (***). Scotopic b-wave: P360: *p* = 0.0001 (***) P500: *p* < 0.00001 (****). Photopic b-wave: P360: *p* = 0.0079 (**) P500: *p* = 0.0292 (*). *N* values: Scotopic a-wave: WT P30: 18; P180: 11; P270: 9; P360: 17; P500: 5. *Ush2a^{delG/delG}* P30: 15; P180: 17; P270: 9; P360: 13; P500:12. Scotopic b-wave: WT P30: 18; P180: 11; P270: 9; P360: 17; P500:5. *Ush2a^{delG/delG}* P30: 15; P180: 17; P270: 9; P360: 12; P500:12. Photopic b-wave: WT P30: 18; P180: 10; P270: 9; P360: 17; P500: 5. *Ush2a^{delG/delG}* P30: 13; P180: 17; P270: 9; P360: 12; P500:12.

indicating that at least a portion of the mutant protein is no longer able to bind to the photoreceptor membrane. Similar fractions from WT retinas probed for FLAG did not detect the presence of mutant USH2A (Fig. 1J).

Sequence prediction suggests that the truncated mutant protein should be approximately 85 kDa in size; however, we consistently observe it around 110 kDa on IB using an anti-FLAG antibody (Fig. 1B). The numerous extracellular domains of USH2A are predicted to be glycosylated[8,30], so to understand whether the mutant protein is glycosylated and whether that accounted for the apparent size difference, retinal extracts from P30 *Ush2a^{delG/delG}* mice were treated with a deglycosylation mixture of enzymes consisting of PNGase F, Neuraminidase, β-*N*-Acetylglucosaminidase, β1-4 Galactosidase, and *O*-Glycosidase to eliminate both N- and O-linked glycosylation. Blots were probed with anti-FLAG and anti-PRPH2 antibodies (the glycosylated protein PRPH2 was used as a positive control). We observed a clear shift of the mutant usherin to the lower size after deglycosylation (Fig. 1L, left image), indicating that it is glycosylated.

### *Ush2a^{delG/delG}* mice exhibit retinal degeneration and mislocalization of opsins

To compare the phenotype displayed by the *Ush2a^{delG/delG}* mice to that exhibited by USH2A patients carrying the c.2299delG mutation, visual acuity was assessed by measuring optomotor response (OMR) in WT

and *Ush2a^{delG/delG}* mice at P45, P180, and P500. *Ush2a^{delG/delG}* mice displayed a statistically significant decrease (**\**p* < 0.01) in visual acuity at P180 (13% reduction) compared to WT mice (Fig. 2A). This defect persisted at P500 at which point we observed a 15.4% reduction when comparing *Ush2a^{delG/delG}* to age-matched WT mice (\**p* < 0.05) (Fig. 2A). We next analyzed retinal function in response to light stimulation by full-field electroretinogram (ERG). Scotopic ERGs at different flash intensities between 0.0045 cd s/m² and 157 cd s/m² were performed and showed no differences in a- and b-wave thresholds between WT and *Ush2a^{delG/delG}* responses (Fig. 2B). Then, we performed scotopic ERGs at P30, P180, P270, P360, and P500 using a single flash intensity of 157 cd s/m² to determine the effect of aging on the ERG responses in the *Ush2a^{delG/delG}* mice. *Ush2a^{delG/delG}* mice exhibited a significant reduction in both scotopic a-wave (28.5% reduction compared to WT, \*\*\**p* < 0.001, Fig. 2C, E) and b-wave (27.2% reduction compared to WT, \*\*\**p* < 0.001, Fig. 2C, F) amplitudes at P360. This decline persisted at P500 when scotopic a- and b-wave amplitudes fell by 30.4% (\*\*\**p* < 0.001) and 35.3% (\*\*\*\**p* < 0.0001) in *Ush2a^{delG/delG}* compared to WT (Fig. 2E, F). The cone response was also decreased in *Ush2a^{delG/delG}* as shown by a significant drop in the amplitude of the photopic b-wave at P360 (21.7% reduction compared to WT, \*\**p* < 0.01) and at P500 (30.2% reduction compared to WT, \**p* < 0.05) (Fig. 2D, G).

To determine whether the scotopic functional decline resulted from a reduction in the number of photoreceptors (rods and cones),

morphometric analysis was performed on retinal sections taken through the optic nerve of P180, P360, and P500 $Ush2a^{delG/delG}$ and WT eyes. We observed a statistically significant reduction (*$p < 0.05$) in the number of photoreceptor nuclei starting at P360 in $Ush2a^{delG/delG}$ mice, a difference that persists at P500 (Fig. 3A, B). The reduction in the number of photoreceptors was associated with perinuclear accumulation of rod (Fig. 3C) and cone (see below Fig. 4) opsins. Accumulation of rhodopsin became obvious as early as P60 in $Ush2a^{delG/delG}$ retinas and increased at P200 and P500 (Fig. 3C, examples highlighted by arrows).

To conclude whether the reduced photopic responses are the result of cone loss, retinal flat mounts from WT and $Ush2a^{delG/delG}$ mice at P500 were labeled either with peanut agglutinin (PNA) or a mixture of antibodies against S- and M-opsins. Four $0.016 \, mm^2$ images for each eye (one in each quadrant) were captured at the central retina, $300 \, \mu m$ away from the optic nerve. The total number of cones labeled either with PNA or cone-opsin antibody from all four areas/eye in a total of three eyes per genotype were averaged (Fig. 4A). Consistent with the decline in the photopic function, the $Ush2a^{delG/delG}$ retina at P500 has a significantly reduced number of cones (14.2%, *$p < 0.05$) when compared to age-matched WT. In addition to the cone loss, we observed pronounced mislocalization of cone opsins to the inner segment and cone terminals (Fig. 4B, arrows) in P500 $Ush2a^{delG/delG}$ retina. To quantify this observation, we counted the number of cones with opsin in the IS or synaptic layer, and found that a statistically significant number of cones exhibit mislocalized opsin in the P500 $Ush2a^{delG/delG}$ retina while mislocalization was absent in the WT retina (Fig. 4C).

To differentiate between the mislocalization of S- and M-opsin, we stained retinal sections from WT and $Ush2a^{delG/delG}$ with PNA and either M- or S-opsin antibody. As shown in Fig. 4D, at P60 $Ush2a^{delG/delG}$ retinas did not show mislocalization of either cone opsin. However, by P200 mislocalization of S- and M-opsin becomes apparent in the $Ush2a^{delG/delG}$ retina and is significantly increased at P500 (Fig. 4B, D). While M- and S-opsin mislocalization is evident, counting the number of cones with mislocalized M- vs, S-opsin revealed that M-opsin mislocalization in the $Ush2a^{delG/delG}$ was far more frequent at both P200 and P500 than S-opsin mislocalization (Supplementary Fig. 2A and Fig. 4D).

Since we observed lower visual capabilities as well as signs of retinal degeneration in the $Ush2a^{delG/delG}$ retinas, we next asked whether the KI model exhibited any signs of cellular stress either accompanying or preceding the visual loss. Upregulation of glial fibrillary acidic protein (GFAP) is a well-established marker for cellular stress and gliosis[31,32] often occurs before or concurrent with retinal degeneration in other mouse models of retinal diseases[33,34] and in $Ush2a^{-/-}$ mice[21]. While P30 and P180 $Ush2a^{delG/delG}$ retinas showed an absence of GFAP upregulation, there were clear signs of gliosis at P360 in the KI retina as shown by the increased levels of GFAP (Fig. 4E).

### Development of ciliary morphologic changes in mutant mice

Since usherin is normally localized to the periciliary region of photoreceptors, we next assessed if the structure of the photoreceptor cilium is affected in the $Ush2a^{delG/delG}$ retina. We measured the length of the CC in the WT and $Ush2a^{delG/delG}$ retina via immunofluorescence after labeling with an anti-centrin 2 (Cen2) antibody[35]. We observed a significant shortening of the CC length as early as P30 in the $Ush2a^{delG/delG}$ retina ($1.57 \pm 0.35 \, \mu m$ for WT and $1.30 \pm 0.30 \, \mu m$ for $Ush2a^{delG/delG}$, measured from 359 cilia from three separate animals for each genotype, error as SD, Fig. 5A, B). The shortening of the CC did not resolve with time and differences persisted at P300 in $Ush2a^{delG/delG}$ retinas ($1.46 \pm 0.32 \, \mu m$ for WT and $1.20 \pm 0.28 \, \mu m$ for $Ush2a^{delG/delG}$, error as SD, Fig. 5C). To confirm our observation, we measured CC length in tissues labeled with anti-acetylated-α-tubulin (acTub) antibody. In line with the results obtained with anti-Cen2 antibody, P30 $Ush2a^{delG/delG}$ retinas labeled with anti-acTub antibody also showed a statistically significant

reduction in cilia length ($1.81 \pm 0.32 \, \mu m$ for WT and $1.66 \pm 0.35 \, \mu m$ for $Ush2a^{delG/delG}$, measured from 177 and 175 cilia from three separate WT and $Ush2a^{delG/delG}$ animals, respectively, error as SD, Fig. 5B). We also analyzed whether ciliary shortening was evident in $Ush2a^{delG/+}$ retinas, consistent with the recessive inheritance of USH2A, no significant changes in ciliary lengths were observed in the heterozygous mice ($1.57 \pm 0.35 \, \mu m$ for WT and $1.51 \pm 0.35 \, \mu m$ for $Ush2a^{delG/+}$, measured from 359 cilia of three separate animals for WT and 222 cilia from two separate animals for $Ush2a^{delG/+}$). The early onset shortening of the CC which precedes photoreceptor degeneration and visual loss strongly suggests the mutant protein causes a developmental defect in the structure of the CC which may contribute to subsequent photoreceptor dysfunction and degeneration.

To further evaluate photoreceptor and CC structure, we performed transmission electron microscopy at P360. Photoreceptor inner segment ultrastructures were normal in $Ush2a^{delG/delG}$ retinas (Fig. 5D). Next, we compared the cross-sectional diameter/width of the OS (OSW) and CC (CCW) in $Ush2a^{delG/delG}$ and WT retinas at P360 (Fig. 5D–F). Interestingly, we observed a significant decrease in the width of the OS in the $Ush2a^{delG/delG}$ retina (13.5% decrease, ****$p < 0.0001$, Fig. 5F), while the width of the CC (Fig. 5F), as well as the length of the OS, were not impacted (Supplementary Fig. 2B). Furthermore, the microtubule number, and arrangement were normal in the $Ush2a^{delG/delG}$ retina (Fig. 5G).

### VLGR1 and WHRN are mislocalized in the $Ush2a^{delG/delG}$ retina

Usherin and VLGR1 are normally part of the periciliary complex in photoreceptor cells[17], and biochemical studies have shown that usherin, VLGR1, and WHRN interact with each other via the usherin PDZ domain to form the USH2 interactome[12]. The PDZ motif is absent in the c.2290delG mutant protein, so it is logical to investigate the localization of USH2A binding partners in the mutant retina. To analyze the localization of these usherin interactors, IF was performed on P30 WT and $Ush2a^{delG/delG}$ retinal sections. In the WT retina, VLGR1 and usherin co-localize with each other at the base of the cilia (Fig. 6A). Similarly, WHRN and VLGR1 co-localize with each other at the base of the cilia in the WT photoreceptor (Fig. 6B), a finding in line with previous studies[12,15,17].

However, the localization patterns of VLGR1/WHRN are profoundly different in the $Ush2a^{delG/delG}$ retina. While in the WT retina, VLGR1 was localized adjacent to the ciliary base and at the outer plexiform layer (OPL, Fig. 6C), in the $Ush2a^{delG/delG}$ retina, VLGR1 was no longer localized at the ciliary base. Instead, VLGR1 accumulated in the inner segment (Fig. 6D, observe lack of alignment with acTub labeling). In the WT retina, WHRN is mainly localized at the ciliary base and some at the OPL (Fig. 6E). Like VLGR1, WHRN loses its localization at the base of the cilia in the $Ush2a^{delG/delG}$ retina and mislocalizes to the inner segment (Fig. 6F). WHRN also exhibits abnormal localization to the proximal part of the OS in the $Ush2a^{delG/delG}$ retina (Fig. 6F, H, examples highlighted by arrows). The aberrant localization of USH2 complex components in the $Ush2a^{delG/delG}$ retina was further highlighted by co-labeling for the mutant USH2A (anti-FLAG) and either VLGR1 or WHRN (Fig. 6G, H). Neither VLGR1 nor WHRN, co-localized with the mutant USH2A protein (Fig. 6G, H), confirming that the interactions between the three key components of the PMC, usherin, VLGR1, and WHRN are disrupted in the $Ush2a^{delG/delG}$ retina. In addition to their localization at the photoreceptor cilium, VLGR1, and WHRN are also present in the OPL in the WT retina (Fig. 6C, E, second image from the left); however, their localization to the OPL was unaltered in the $Ush2a^{delG/delG}$ (Fig. 6D, F, second image from the left).

In addition to the core USH2 complex (WHRN/VLGR1/usherin), other usher proteins such as SANS (USH1G)[15] and harmonin (USH1C or HARM)[25] also participate in the usher interactome at the periciliary area. We thus asked whether these and other usher proteins retained their ciliary localization in the $Ush2a^{delG/delG}$

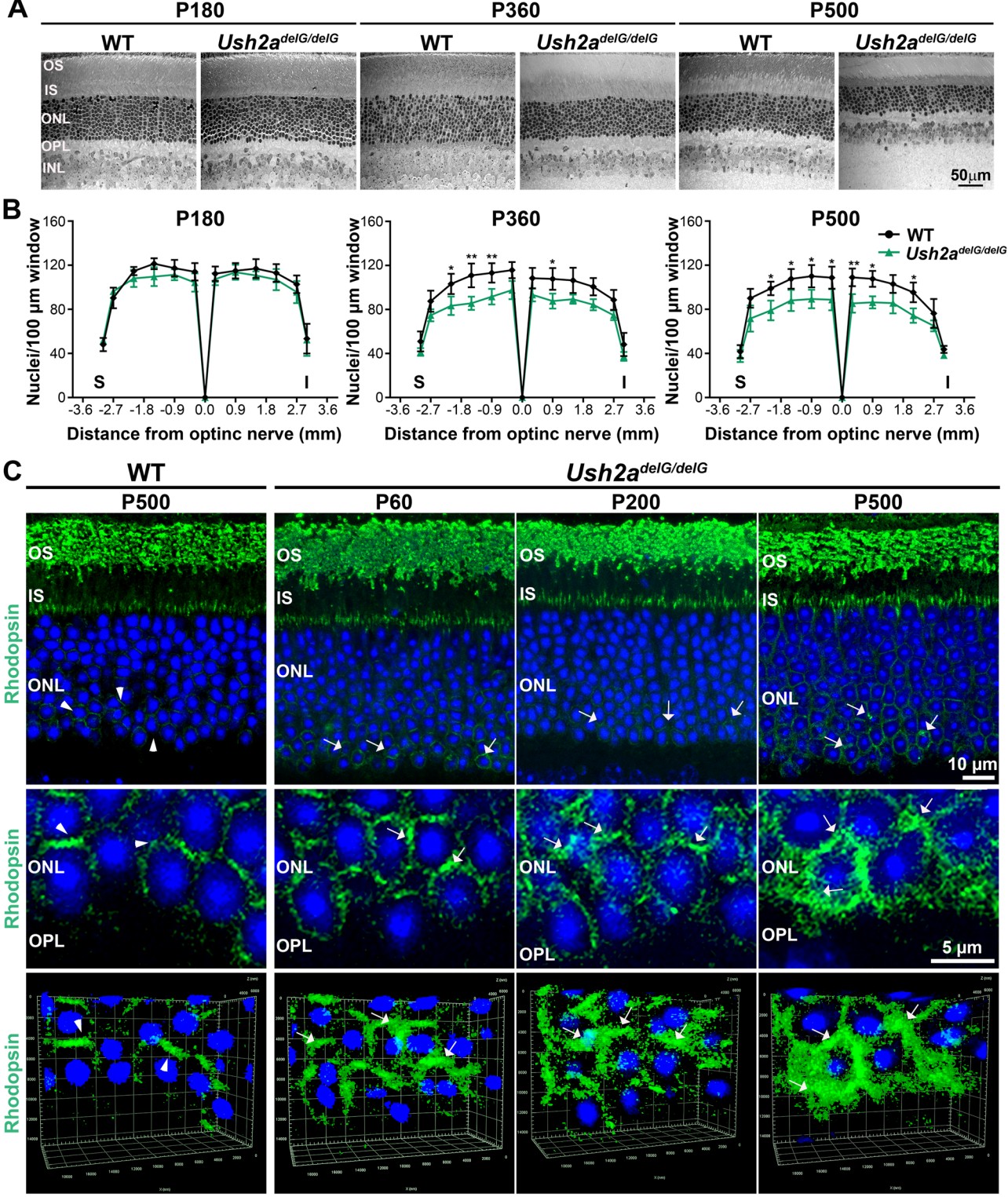

**Fig. 3 | Expression of mutant usherin leads to late-onset photoreceptor degeneration and mislocalization of rhodopsin. A** Representative retinal light micrographs are shown for P180, P360, and P500 *Ush2a^{delG/delG}* and WT mice. **B** The number of photoreceptors in the ONL was determined by counting the number of nuclei at the indicated distances from the optic nerve and data are presented as means ± SEM. *N* values: 4 for WT P360, 3 for all other tested genotypes and time points. A significant decrease in the ONL thickness is observed at P360 persisting at P500. Significance was determined by two-way ANOVA with a Bonferroni post hoc test. *P* values: P360: Superior: −2.1 mm: *p* = 0.0222 (*), −1.5 mm: *p* = 0.0012 (**), −0.9 mm: *p* = 0.0095 (**). Inferior: 0.9 mm: *p* = 0.0204 (*). P500: Superior: −2.1 mm:

*p* = 0.0420 (*), −1.5 mm: *p* = 0.00486 (*), −0.9 mm: *p* = 0.0313 (*), −0.3 mm: *p* = 0.0420 (*). Inferior: 0.3 mm: *p* = 0.0078 (**), 0.9 mm: *p* = 0.0232 (*), 2.1 mm: *p* = 0.0270 (*). **C** Retinal sections from WT (P500) and KI animals at P60, P200, and P500 were labeled with an antibody against rhodopsin. The mislocalization of rhodopsin is observed in the *Ush2a^{delG/delG}* retinas. Shown are single-plane images with lower images taken at higher magnifications. White arrowheads highlight the weak basic rhodopsin signal located in the ONL in P500 WT. White arrows point to mislocalized rhodopsin. OS outer segments, IS inner segments, ONL outer nuclear layer, OPL outer plexiform layer, INL inner nuclear layer, S superior, I inferior.

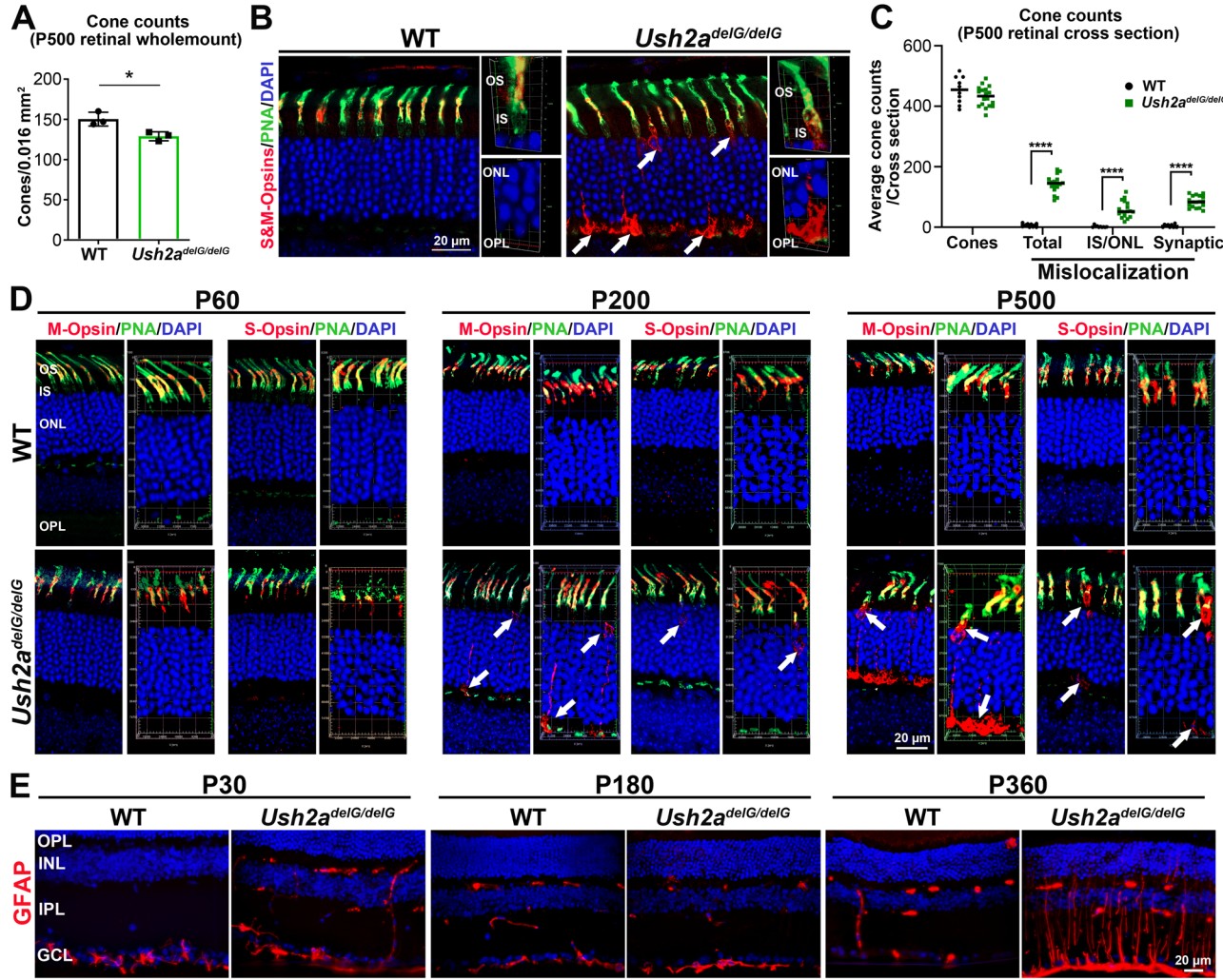

**Fig. 4 | The *Ush2a^{delG/delG}* retina shows a decline in cone number, mislocalization of cone opsins, and an upregulation of the stress marker GFAP. A** Cone cell density count from four 0.016 mm² areas per retinal whole mount from P500 *Ush2a^{delG/delG}* and WT mice (one in each petal) taken at a 300 μm distance from the optic nerve (see the Method Section) and stained with PNA, showing that number of cones is significantly reduced in the *Ush2a^{delG/delG}* retina mirroring the functional decline. Shown are means ± SEM from *N* = 3 animals. *P* value: *p* = 0.0221 (*) by a two-tailed unpaired *t*-test. **B** Retinal sections from the indicated genotypes at P500 were co-labeled with PNA (green) and a mixture of anti-S- and M-opsins (red) antibodies. Nuclei in all sections were counterstained with DAPI (blue). White arrows show an accumulation of cone opsins in the ONL and IS. Two panels on the right side of each genotype showing 3D construction of the OS/IS and ONL/OPL areas where mis-localization of cone opsins is observed. Shown are representatives of *N* = 3 animals per genotype. **C** Total number of cones counted from the entire retinal cross sections shown in **B**. Also shown are the number of cones with mislocalized cone-

opsins either in the IS/ONL or at the synaptic terminals. *N* values: WT: Presented are average cone counts from two sections per eye from five independent mice. *Ush2a^{delG/delG}*: Presented are average cone counts from three sections per eye from six independent mice. Significance was determined by a two-way ANOVA with a Bonferroni post hoc test. *P* values: total: *p* < 0.0001 (****), IS/ONL: *p* < 0.0001 (****), synaptic: *p* < 0.0001 (****). **D** Paraffin retinal sections from WT and *Ush2a^{delG/delG}* at P60, P200, and P500 were labeled for PNA (green) and immunolabeled either for S- or M-opsin (red) and counterstained with DAPI (blue). On the right side of each panel a 3D reconstruction of 8-12 confocal stacks (1 μm thickness) is shown, high-lighting the photoreceptor cell layer. White arrows pointing to S- or M-opsin mis-localized in the IS, ONL, or synaptic terminal. **E** Retinal sections from WT and KI animals at P30, P180, and P360 were labeled with antibodies against GFAP. At P360, upregulation of GFAP was observed. Shown are single-plane images. OS outer segments, IS inner segments, ONL outer nuclear layer, OPL outer plexiform layer, INL inner nuclear layer, IPL inner plexiform layer, GCL ganglion cell layer.

retina. Co-labeling for HARM and acTub (Fig. 7A) or SANS and acTub (Fig. 7B) confirmed that these two proteins localize to the base of the CC in the WT retina and showed that their localization was retained in the *Ush2a^{delG/delG}* retina. We also evaluated the localization of Cadherin 23 (CDH23, USH1D), Protocadherin 15 (PCDH15, USH1F), calcium and integrin binding family member 2 (CIB2, USH1J) and Clarin1 (CLRN1, USH3A). As with HARM and SANS, none of these USH proteins displayed altered localization in the *Ush2a^{delG/delG}* mutant retina (Fig. 7C−F). These data highlight the critical role of usherin in the formation of properly localized USH2 complexes with WHRN and VLGR1, but indicate that the localiza-tion of other USH proteins does not depend on the PDZ binding domain of usherin.

## Heterozygous mutant mice exhibit normal retinal function and structure

In line with the recessive inheritance of USH2A, no significant change in visual responses at P60, P150, P400, and P550 by ERG were observed in heterozygous *Ush2a^{delG/+}* mice. Neither scotopic a- and b-wave, nor photopic b-wave were significantly affected at any of the time points up to P400 (Supplementary Fig. 3A). Although there is a declining trend in scotopic and photopic ERG responses at P550, it did not reach statistical significance (Supplementary Fig. 3A). Histological analysis at the light levels of the retinas at P180, P360, and P500 also showed no significant structural defects or reduction in the number of photo-receptor nuclei when compared to their aged-matched WT controls (Supplementary Fig. 3B).

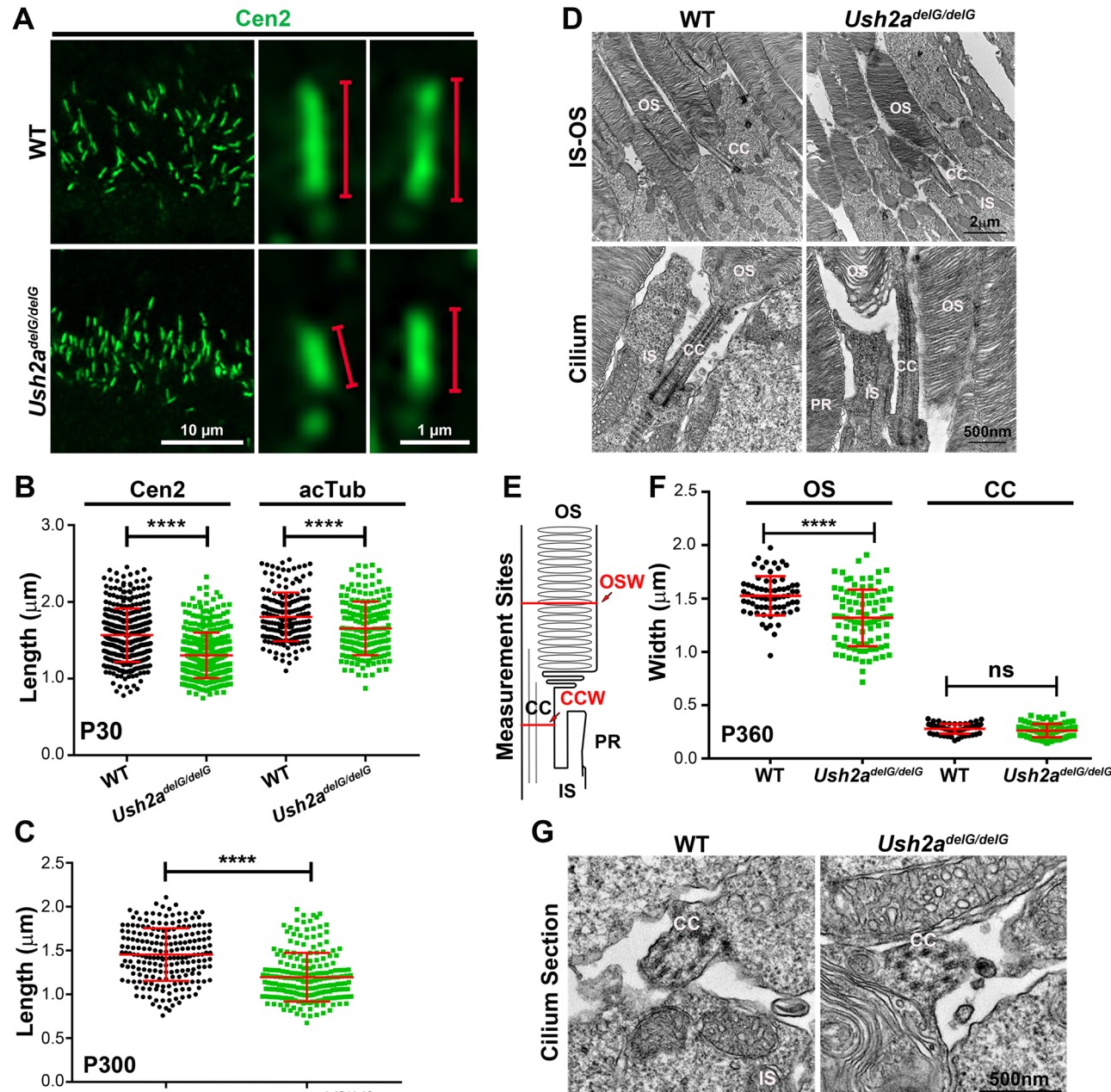

**Fig. 5 | Expression of mutant usherin leads to shortening of the photoreceptor cilium. A** Retinal sections labeled with the CC marker Cen2 revealed a decrease in CC length in the *Ush2a^delG/delG* retina. **B** Measurement of the ciliary length revealed statistically significant shortening at P30 *Ush2a^delG/delG* mice (359 cilia were measured from three independent WT and *Ush2a^delG/delG* mice). The acetylated region of the photoreceptor cilium is also significantly reduced (177 and 175 cilia were measured from three separate WT and *Ush2a^delG/delG* mice, respectively). Significance was determined by a two-tailed unpaired *t*-test, $p < 0.0001$ (****). **C** Measurements of the ciliary length revealed the shortening to be significant in P300 *Ush2a^delG/delG* mice (208 and 210 cilia were measured from two separate WT and *Ush2a^delG/delG* mice, respectively). Significance was determined by a two-tailed unpaired *t*-test, $p < 0.0001$ (****). Data presented in **B** and **C** are means ± SD. **D** Representative TEM

images of the OS/IS junction (upper images) and higher magnification images of the CC (lower images) from P360 WT and *Ush2a^delG/delG* mice are shown. **E** A photoreceptor schematic denoting the areas at which measurements of OS and CC widths were made. **F** The width of OS and CC was quantified in P360 WT and *Ush2a^delG/delG* retinas. The width of the OS was significantly reduced in the *Ush2a^delG/delG* retina, while the width of CC remained unchanged (68 cilia were measured for WT and 90 cilia were measured for the *Ush2a^delG/delG*). Shown are means ± SD. Significance was determined by a two-tailed unpaired *t*-test, $p < 0.0001$ (****) for OS width. **G** High magnification TEM images are taken from P360 WT and *Ush2a^delG/delG* mice showing a cross-section of the CC. OS outer segment, CC connecting cilium, IS inner segment, PR periciliary region, OSW outer segment width, CCW connecting cilium width.

Since we observed profound mislocalization of c.2290delG mutant usherin accompanied by mislocalization of the other USH2 proteins, VLGR1 and WHRN, in the *Ush2a^delG/delG*, we next asked whether the presence of fifty percent of WT usherin could improve this phenotype. *Ush2a^delG/+* heterozygous retinal sections were co-labeled for USH2 proteins. In the P30 *Ush2a^delG/+* retina, WT usherin is normally

localized to the periciliary ridge together with anti-acTub labeling and confined to the base of the photoreceptor cilium and at the photoreceptor synapses in the OPL (Fig. 8A, B). Critically, co-labeling for VLGR1 with either acTub or WT usherin showed that VLGR1 co-localized with WT usherin at the base of the CC in the *Ush2a^delG/+* retina (Fig. 8C, D), in line with its localization observed in the WT retina.

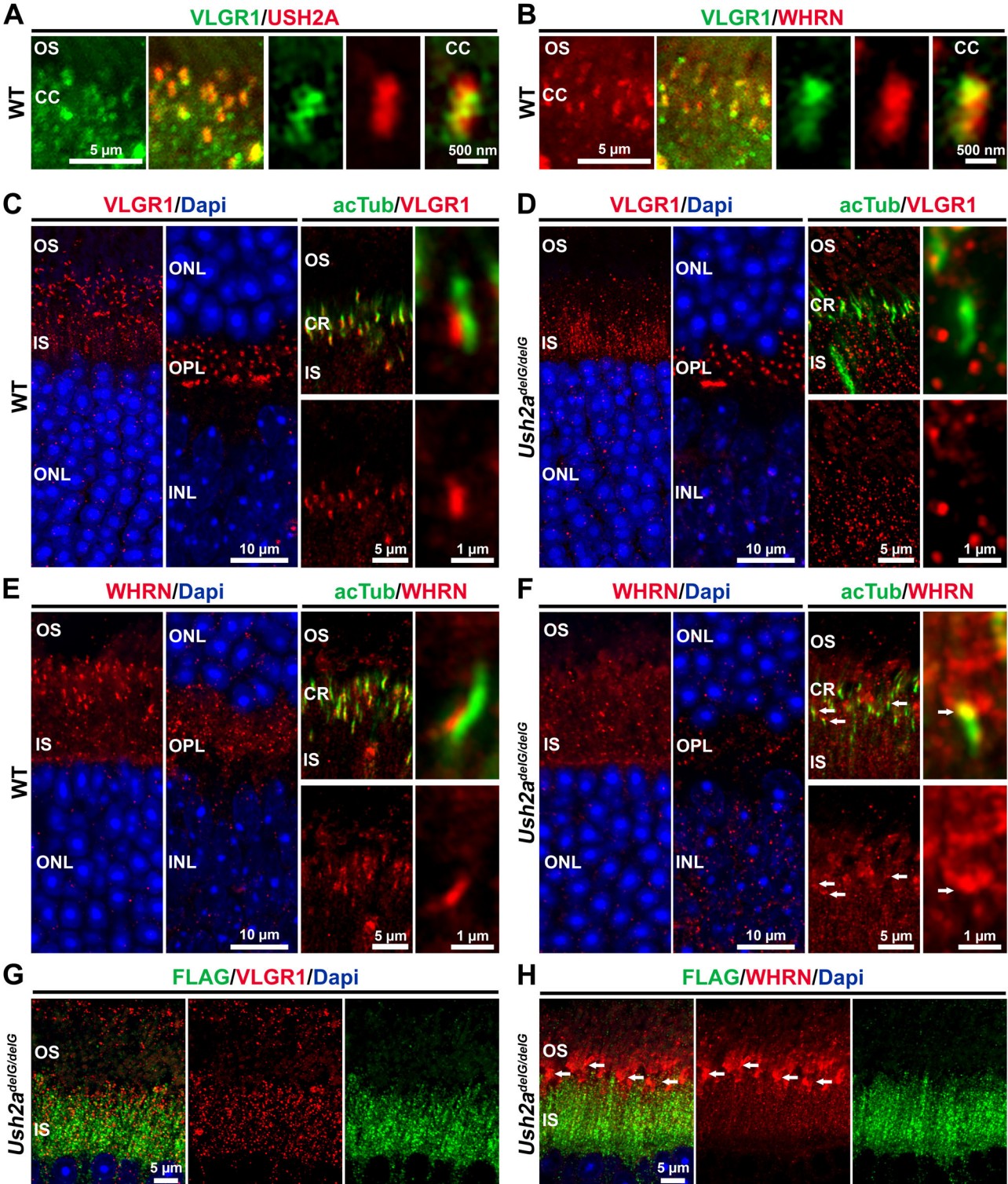

**Fig. 6 | VLGR1 and WHRN are mislocalized in the *Ush2a^delG/delG* retinas. P30 WT and *Ush2a^delG/delG* retinas were used.** VLGR1 co-localizes with usherin (**A**) and WHRN (**B**) adjacent to the ciliary base in WT retinas. **C** VLGR1 localizes between the IS and OS and in the OPL (left two images). VLGR1 is localized in the ciliary region (second image from the right) and rightmost image shows a high-magnification image demonstrating VLGR1 localization at the ciliary base. **D** VLGR1 is mislocalized to the IS (leftmost image) while its localization in OPL is maintained (second image from the left) in *Ush2a^delG/delG*. Co-staining for acTub shows the absence of VLGR1 from the CC (right images). **E** WHRN is localized between the IS and OS, and at the OPL in the WT retina (left image). Co-labeling for acTub and WHRN demonstrates the localization of WHRN at the CR (second image from the right). A high-magnification image of a single photoreceptor cilium demonstrates WHRN

localizing at the ciliary base (rightmost image). **F** WHRN is mislocalized to the proximal part of the OS in the *Ush2a^delG/delG* retina (left panels). Co-staining for acTub and WHRN revealed the absence of WHRN from the CC (second image from the right) as well as the mislocalization to the proximal part of the OS (examples highlighted by arrows). A high magnification image demonstrates the absence of WHRN from the CC, and its mislocalization towards the proximal OS (rightmost image, highlighted by arrow). **G** VLGR1 and FLAG are independently localized in IS in the *Ush2a^delG/delG* retina. **H** WHRN in the *Ush2a^delG/delG* retina is localized in the OS, while the truncated usherin is localized in IS of *Ush2a^delG/delG* retina. OS outer segment, CR ciliary region, CC connecting cilium, IS inner segment, ONL outer nuclear layer, OPL outer plexiform layer, INL inner nuclear layer.

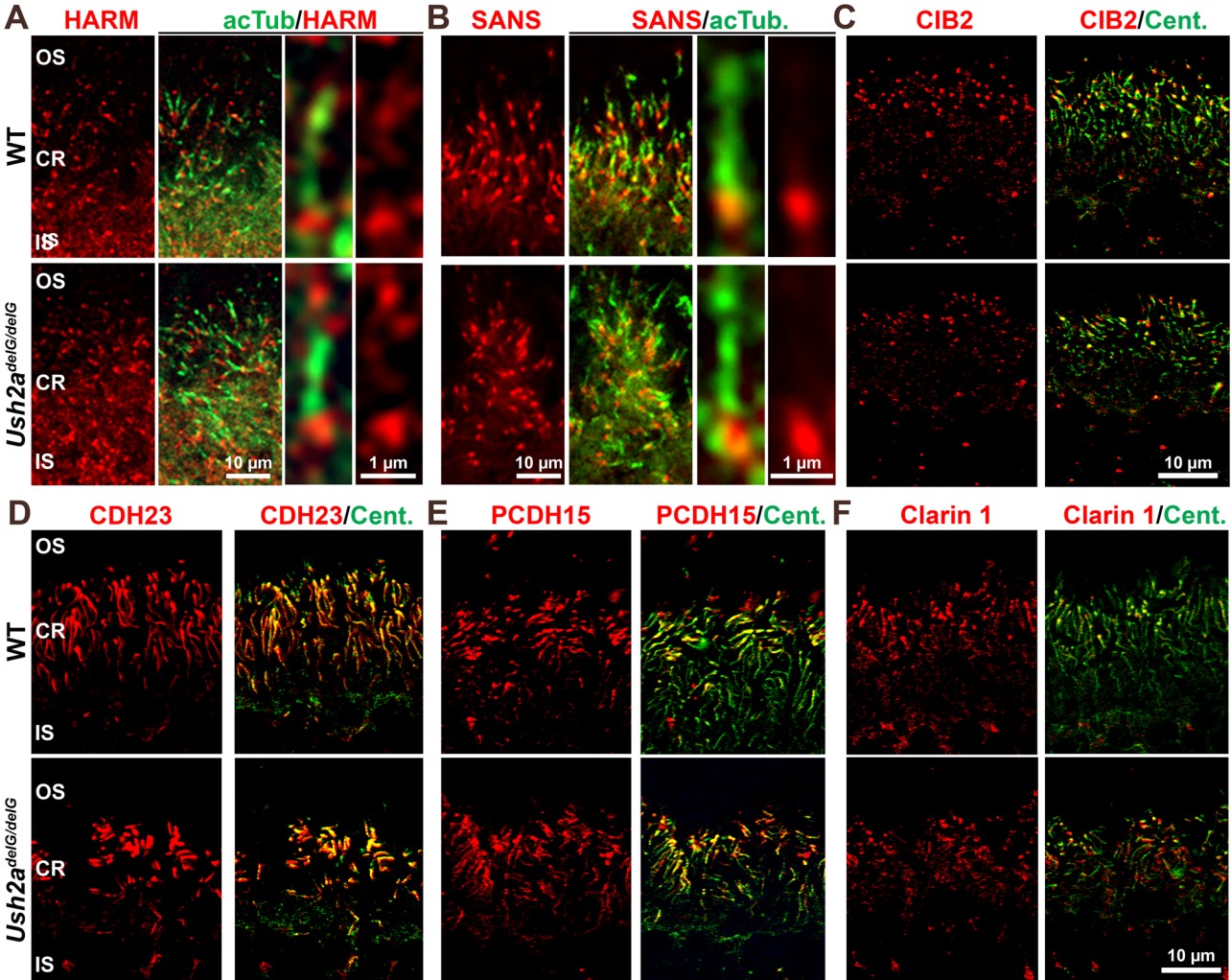

**Fig. 7 | Proper localization of other USH proteins in the *Ush2a^delG/delG*^ retinas. A–F** P30 retinal sections from WT and *Ush2a^delG/delG*^ mice were labeled with antibodies against other USH proteins as listed above. The localization of all proteins tested was unaltered in the *Ush2a^delG/delG*^ retina when compared to WT. OS outer segment, CR ciliary region, IS inner segment.

Similarly, WHRN localization at the base of the CC is preserved in the *Ush2a^delG/+^* retina, and it is still co-localized with VLGR1 (Fig. 8E, F). In contrast, the expression of fifty percent WT usherin does not completely prevent the mislocalization of the c.2290delG mutant usherin. We observed cilia in the *Ush2a^delG/+^* retina in which WT and mutant usherin (FLAG) co-localized (Fig. 8H, I) but, as in the *Ush2a^delG/delG*^ photoreceptor, a large portion of the c.2290delG mutant usherin accumulated in the inner segment in the *Ush2a^delG/+^* photoreceptor (Fig. 8G, H for *Ush2a^delG/delG*^ and *Ush2a^delG/+^*, respectively).

Since we observed the c.2290delG mutant usherin to be partially localized at the PMC in the *Ush2a^delG/+^* retina, we next asked whether this was also evident biochemically. We performed sub-cellular fractionation of retinal extracts from P30 *Ush2a^delG/+^* retinas and found that WT usherin in the *Ush2a^delG/+^* retina exclusively localizes to the membrane fraction similar to its localization in the WT retinas, indicating that the presence of the mutant usherin in the *Ush2a^delG/+^* retina does not interfere with the ability of the full-length usherin to bind to the membranes (Supplementary Fig. 4). The c.2290delG mutant usherin, on the other hand, was found in the cytosolic and membrane fractions in the *Ush2a^delG/+^* retina (Fig. 8J) comparable to the results obtained for the *Ush2a^delG/delG*^ retinas (Fig. 1K). The presence of the truncated protein in the membrane fraction was an unexpected result, given the lack of a transmembrane domain. Given that most of the truncated protein is localized in the inner segments of the *Ush2a^delG/+^* retina (Fig. 8H), we

assessed whether it is trapped in the endoplasmic reticulum (ER). Retinal sections from P30 *Ush2a^delG/delG*^ and *Ush2a^delG/+^* mice were co-labeled with anti-FLAG and the ER marker calreticulin. In the *Ush2a^delG/delG*^ retina, we observed co-localization of FLAG and calreticulin (Fig. 8K, white arrows in right upper panel) while the co-localization was less prominent in the *Ush2a^delG/+^* retina (Fig. 8K, right upper panel). As seen earlier (Fig. 8H, I), we detect a portion of the c.2290delG mutant usherin at the periciliary area in the *Ush2a^delG/+^* retina (Fig. 8K).

## Discussion

Establishing a mouse model for USH2A that reiterate patient's phenotypes has been an important goal for the future development of therapeutic strategies. Several mouse models for USH-linked genes have been successful in recapitulating the auditory and vestibular phenotypes but not the retinal phenotype (for review see refs. [3,4]), perhaps due to gene deletions rather than knocking-in a disease mutation. While several of these mouse lines display hearing defects allowing the study of the auditory phenotype of USH, studies on the retina are needed to establish causative effects before any attempts for therapeutic interventions. Thus, a genetic mouse model that can recapitulate USH2A retinal phenotype was the goal of the current study. In this work, we generated a knock-in mouse model for one of the most common mutations in USH2A (c.2299delG) that exhibited a retinal phenotype comparable to USH2A patients with the c.2299delG

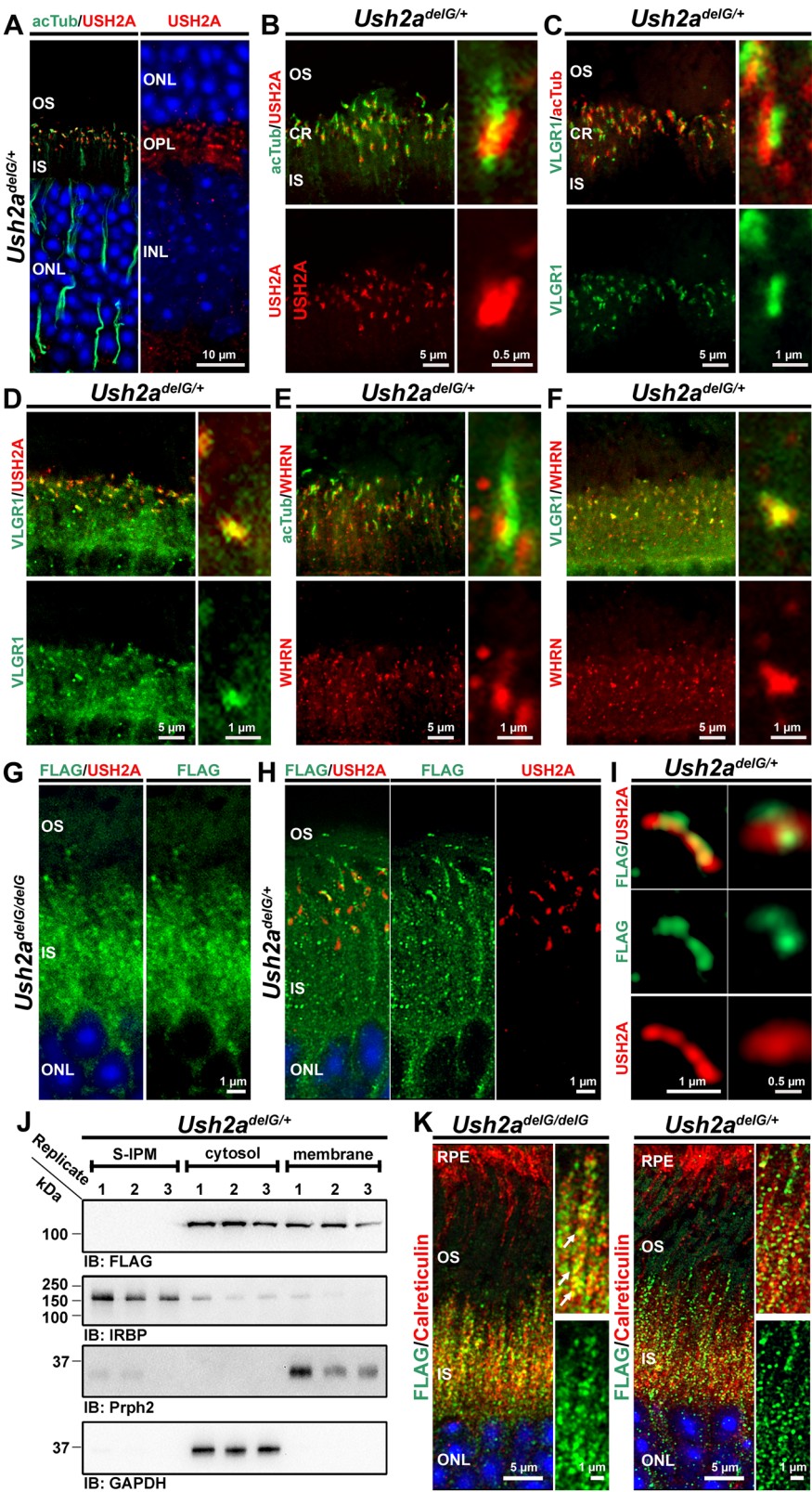

mutation. While WT usherin is found at the PMC alongside the CC and at the photoreceptor synaptic terminal[15,17,24,26], truncated usherin did not exhibit this restricted localization at the CC, rather it was intracellularly mislocalized throughout the inner segment. Biochemical analysis of the truncated protein revealed a band at ~110 kDa in size that was found to be glycosylated, and lost its ability to interact with other proteins covalently or non-covalently. Furthermore,

mislocalization of mutant usherin to the inner segment led to mislocalization of its interacting partners, VLGR1 and WHRN, which are necessary for periciliary complex formation. Our findings also identified an important role for usherin in regulating the structure of the periciliary ridge since we observed a shortening of the photoreceptor connecting cilium and mislocalization of the visual pigments in the KI retina.

**Fig. 8 | Preservation of periciliary localization of VLGR1 and WHRN in the Ush2a[delG/+] retina. A, B** Labeling of the retinal section from P30 Ush2a[delG/+] mice for usherin (USH2A) and acTub demonstrates its localization at the periciliary base (**A, B**) and the OPL (**A**, right image) in Ush2a[delG/+] mice. **C, D** Labeling for VLGR1 and acTub revealed its localization at the ciliary base (**C**), where VLGR1 co-localizes with usherin (**D**) in the Ush2a[delG/+] retina. **E, F** Staining for WHRN demonstrates its localization at the ciliary base, where it co-localizes with VLGR1 (**F**) in the Ush2a[delG/+] retina. **G, H** Staining for FLAG in Ush2a[delG/delG] (**G**) and Ush2a[delG/+] (**H**) retinas revealed a portion of the truncated usherin (FLAG) to be localized at the photoreceptor cilium in the Ush2a[delG/+] retina but not in Ush2a[delG/delG]. **I** High magnification images of single photoreceptor cilia demonstrating the co-localization of truncated and full-length usherin at the ciliary base in the Ush2a[delG/+] retina. **J** Immunoblot of extracts from cellular fractionation of Ush2a[delG/+] retinas demonstrated that mutant usherin is found in the cytosol and membrane fractions. **K** Co-labeling of retinal sections from P30 Ush2a[delG/delG] (left images) and Ush2a[delG/+] (right images) for truncated usherin (FLAG) and the ER marker calreticulin. The truncated protein showed a prominent accumulation at the ER in the Ush2a[delG/delG] retina, while in the Ush2a[delG/+] retina this accumulation was less pronounced. OS outer segment, CR ciliary region, IS inner segment, ONL outer nuclear layer, OPL outer plexiform layer, INL inner nuclear layer.

The c.2299delG mutation in *USH2A* can lead to aberrant splicing and exon skipping in some patient transcripts which may result in additional potential truncated protein products[36]. Nasal samples from patients with c.2299delG-associated USH2A exhibited variable USH2A transcript profiles including many transcripts with exons 13 and/or 12 already skipped[36]. Bioinformatic analysis suggested that the c.2299delG change likely disrupted an exonic splicing enhancer and created an exonic splicing silencer within exon 13 that led to the skipping of exons 12 and 13 to varying degrees across samples. Although the deletion of exon 13 yields an in-frame message, the deletion of exons 12 and 13 yields an out-of-frame message predicted to generate a truncated protein. While results from RNA-based therapeutic approaches[37] and exon-skipping strategies are eagerly awaited, their effect on the phenotype remains to be determined, and significant further information is needed about the various isoforms of usherin expressed in the retina from both normal and c.2299delG alleles. To understand the pathogenic defects underlying this mutation, it is essential to generate an animal model for it as we have done by introducing the c.2290delG mutation (comparable to the human c.2299delG) along with the 20 amino acid human extension in the mouse *Ush2a* gene to study the functional effect of mutant usherin. Nevertheless, it is still unclear how missing or truncated usherin leads to hearing/vision impairments in patients. It is also unclear why some *USH2A* gene mutations result in USH2, while others cause RP without hearing loss[5,38–40]. In the sensory hair cells of the cochlea, USH2A is transiently associated with the hair bundles during postnatal development, suggesting an important role in their maturation[41]. In photoreceptors, USH2A is localized to a spatially restricted membrane microdomain at the PMC, suggesting a role in cargo transport between the inner and outer segment[17]. This role is confirmed in our KI model where both of rod and cone opsins were found mislocalized.

Interestingly, the absence of usherin in *Ush2a[−/−]* seems to have much less effect on the retina than the presence of the truncated protein. The early onset retinal changes observed in our KI model include shortening of the CC and mislocalization of the core PMC components of mutant usherin, VLGR1, and WHRN. These changes manifest as early as P30. While the ciliary structure was not analyzed in *Ush2a[−/−]* mice, a disruption of the PMC in the retina could also be shown in the *Ush2a[−/−]* retina, however the earliest time point at which this disruption was seen was not stated[12]. An additional difference to our model is that in the *Ush2a[−/−]* retina, both WHRN and VLGR1 are depleted rather than mislocalized. A second retinal phenotype displayed by our KI model was the mislocalization of rhodopsin and cone opsins starting at P60 and P200, respectively. Although rhodopsin mislocalization was not reported for the *Ush2a[−/−]* model, cone opsins were mislocalized as early as P80[21,42]. While data on CC structure and localization of opsins and PMC components are not available for patients, a functional decline in the retina in USH2A patients manifests pre- or post-pubescence, usually in the second decade of life[3–5,43]. The first sign of a decline in visual capabilities in the KI mice was observed at P180 (comparable to the second decade of life in patients) with a significant decrease in OMR. ERG responses gradually decreased as the animals got older and reached statistical significance at P360. This is different from USH2 patients, in whom decreased ERG responses

are observed at a very young age[3]. However, when compared to the *Ush2a[−/−]* model which displays decreased ERG responses only at 20 months of age, our KI model performs better in reproducing the retinal phenotype of patients[21]. In line with this, our KI model showed a significant thinning of the ONL as early as P360, while ONL thickness remained unaffected in the *Ush2a[−/−]* model as late as 20 months of age.

The formation of the truncated usherin protein in our KI model and its mislocalization to the inner segment is a key finding. Furthermore, the absence of the truncated protein at the ciliary ridge in our c.2290delG model mimics observations made in the *Ush2a[−/−]* model (i.e., it also has no usherin at the ciliary ridge), but the presence of the c.2290delG mutant usherin in the cytosol is likely what accelerates the onset of retinal degeneration in our model, and suggests that the truncated protein may play a critical role in the time-course and mechanism of disease in patients. In a follow-up work using the *Ush2a[−/−]*[12], the authors reported that the absence of usherin led to the absence of WHRN and VLGR1 from the PMC. In contrast, we find that when truncated usherin is mislocalized from the ciliary ridge to the inner segment, WHRN and VLGR1 also mislocalize to the inner segment (rather than being absent). Combined these findings suggest that usherin and its partners, WHRN and VLGR1, are needed for the proper assembly of the periciliary ridge. Although this assembly is crucial for the long-term health of photoreceptors, disease mechanisms in patients are likely more complex, reflecting both loss-of-function contributions from the lack of appropriately localized and assembled usherin complexes at the ciliary ridge and gain-of-function contributions from the abnormal accumulation of the truncated usherin in the inner segment. The earlier onset of the retinal phenotype observed in the KI model in comparison to the *Ush2a[−/−]* was an expected outcome since many USH2A mutations are proposed to generate truncated proteins.

Inner segment mislocalization of truncated usherin is an interesting observation, as one would expect it to be secreted to the extracellular space due to the presence of the secretory signal at the N-terminus and the absence of its transmembrane domain. Its intracellular trapping is likely in membrane compartments/vesicles since it is glycosylated. This suggests the presence of essential elements within the missing domains in the mutant protein that are required to expel it to the extracellular space and locate it at the periciliary ridge, as well as complexing it with the full-length native usherin and/or with VLGR1 and WHRN. Alternatively, the addition of the 20 amino acids observed in patients at the end of the truncated usherin interferes with its PMC localization or secretion. We also found more of the truncated usherin co-localized with the ER marker calreticulin in the *Ush2a[delG/delG]* retina, likely inducing unfolded protein response and leading to GFAP upregulation and cell death. This accumulation in the ER suggests that the truncation and/or addition of the 20 amino acids results in misfolding of the protein, preventing it from exiting the ER which likely leads to a more prominent phenotype when compared to the *Ush2a[−/−]* retina. Accumulation of misfolded proteins in the ER has been shown to cause cell death in neurodegenerative and inherited retinal diseases[44–47]. In addition to the 20 amino acids, our model also contains a 3x FLAG tag, which represents a difference between the protein expressed in our model, and the mutant protein expressed in patients. However, given

that thus far no evidence of misfolding caused by this commonly used tag has been described, it seems rather unlikely that the ER accumulation observed in the KI mice is caused by the presence of the 3x FLAG tag.

The importance of having a truncated or mutant protein expressed in order to recapitulate the retinal phenotype is supported by the fact that most knockout models of USH proteins fail to display effects in the retina (reviewed in ref. [41]). Apart from the above-described *Ush2a*[-/-] model, only a USH2D model in which the long isoform of whirlin is specifically knocked out displayed a robust retinal phenotype[12]. However, just like in the *Ush2a*[-/-] model this retinal phenotype only manifests in old mice (between 28 and 33 months of age). In line with our KI model and the *Ush2a*[-/-] model, the PMC is also disrupted in this USH2D model. For *Ush3a*[-/-] mice (clarin-1) conflicting results were obtained, with a recent study proving a significant decrease in ERG responses observed as early as P90[48], while an older study could not find any retinal phenotype[49]. However, retinal degeneration was absent in both studies. Studies utilizing mouse models in which the mutant protein is expressed were more successful in reproducing the retinal phenotype. A study analyzing mice with different mutations in the Myo7a (USH1B) allele found a decreased ERG response in several of these mutant models starting as early as P70[50]. A USH1C knockin model expressing mutant HARM (*Ush1c*[c.216>A]) successfully reproduced the retinal phenotype observed in USH1C patients[51]. Here, a significant reduction in ERG responses as well as retinal degeneration could be observed (onset at P30 and P180, respectively). A second knockin with a prominent retinal phenotype was the USH1F model (*Ush1f*[R250X]) expressing a truncated version of PCDH15[40]. This model displayed an early onset reduction in ERG responses at P30 and retinal degeneration between 12–14 months of age.

Further evidence for the importance of having mutated or truncated usherin expressed in order to recapitulate the human disease phenotype is supported by data obtained from USH2A zebrafish models. Several zebrafish models have been generated for USH2A, which, in contrast to mouse models, mainly showed moderate retinal phenotypes with no auditory defects. Among these are *Ush2a*[rmc1], *Ush2a*[b1245], and *Ush2a*[uS07] zebrafish models[52,53]. The frameshift mutation in exon 13 of Ush2a in the *Ush2a*[rmc1] zebrafish completely abolished usherin in the photoreceptors while the targeted frameshift mutation in exon 71 in the *Ush2a*[b1245] model resulted in a truncated protein that lacks the C-terminal 62 intracellular amino acids[52]. This mutant also lacked the PDZ binding domain but was able to locate at the periciliary membrane of the *Ush2a*[b1245] photoreceptors. Interestingly, in both of these zebrafish models WHRN was absent from the periciliary region as was observed in the *Ush2a*[-/-] and our *Ush2a*[delG/delG] KI model[52]. Unlike our model, retinal degeneration in both of these zebrafish models was induced by constant bright illumination for three days[52]. The *Ush2a*[uS07] model which carries a mutation in exon 12 resulting in the introduction of premature translational stop that led to the absence of usherin in the retina and cochlea[53]. The *Ush2a*[uS07] model showed a much milder retinal degeneration that became slightly apparent at 12 months post fertilization (mpf) and predominantly in rods and did not display a significant reduction in the OMR. While the study did not address how the usherin interactors SANS, WHRN, and VLGR1 were affected in the *Ush2a*[uS07] model, like our KI model, a mislocalization of rhodopsin and blue cone opsin towards the inner segments was observed[53] without the mislocalization to the OPL as displayed by the *Ush2a*[delG/delG] retina. Despite the known advantages and disadvantages of both mouse and zebrafish models, when researching the retinal USH phenotype, complemental studies using both models seem to be the most promising approach. Unlike zebrafish models, which have a robust retinal regenerative capacity, the mammalian retina (including humans and mouse) has a very limited capacity to replace lost neurons. This poses limitations in using zebrafish models to understand disease progression as they respond differently to acute retinal degeneration than the human system. Also, the significantly higher percentage of cones in the zebrafish (92% in larvae and 60% in adults) when compared to the human (around 5%) might prove to be a disadvantage in the research of the retinal USH phenotype, considering that loss of rod function precedes that of cones in patients[1,43,54,55].

Another key finding in our study that has not been addressed in other USH2 models are the morphologic changes to the CC at P30, significantly earlier than the functional or structural changes. A connection between the shortening of the photoreceptor cilium and retinal diseases has been previously established[56,57]. In addition, this shortening is likely associated with opsin mislocalization. For example, one recent study where CC-shortening and RP were associated with a KIF3B mutation[58], reported that failure to regulate CC length resulted in inner segment mislocalization of rhodopsin, a phenotype we also observed in the *Ush2a*[delG/delG] model. Another study found that shortening of CC preceded photoreceptor degeneration and rhodopsin mislocalization in the *Rd16* mouse model[59]. However, the timeframe of degeneration in the *Rd16* mouse is much faster than in *Ush2a*[delG/delG], with the CC shortening occurring at P10 followed by photoreceptor degeneration as early as P16. Combined, these studies provide support for the idea that CC dysregulation is a critical cellular contributor to the ensuing functional and structural changes.

It is not surprising that structural CC defects would lead to impaired retinal function since it is the main transport hub of cargo between the inner segment and OS. Usherin was described to play an important role in trafficking cargo, including opsins through the CC[21,42]. Usherin, VLGR1, and WHRN with SANS together form the PMC[12,17,18], a complex that is proposed to be the mammalian equivalent to the amphibian periciliary ridge complex. The latter is a docking site for opsin-containing vesicles in the frog photoreceptor[60–62]. The absence of VLGR1 and WHRN from *Ush2a*[delG/delG] at the CC suggests that the PMC in the photoreceptors is disrupted, upsetting the transfer of opsins from the inner segment most likely by preventing the efficient loading of opsins into the cilium. The inability to properly load a sufficient amount (or at an insufficient rate) of newly synthesized opsin for trafficking to the OS while maintaining the same rate of de novo synthesis would likely result in opsins' mislocalization. This explains the perinuclear accumulation of rhodopsin and inner segment and OPL mislocalization of cone opsins in the *Ush2a*[delG/delG] retinas. Furthermore, this also explains the thinner OSs observed in our model. The importance of the PMC and the correct localization of USH proteins in it for rhodopsin transport is further supported by observations made in the whirler mouse (USH2D model)[20] and a mouse model carrying a mutation in MYO7A (USH1B model)[63]. Both proteins were found to interact with usherin at the ciliary base[12,17] and the disruption of the PMC observed in these mice coincided with rhodopsin mislocalization. However, how the disruption of the PMC mediates the morphologic defects in the cilium is currently not clear. Interestingly, the PMC member SANS and its interaction partner HARM were not mislocalized in the *Ush2a*[delG/delG] retina. HARM is described to function mostly as a scaffold protein organizing the USH interactome[25,26]. While SANS is also described as a scaffold protein, recent studies have found evidence for a multitude of additional functions for SANS, including microtubule-based transport, endocytosis, ciliogenesis, and splicing[13,15,64,65]. The localization of HARM and SANS was retained in the *Ush2a*[delG/delG] retina, proving that the organization of usherin, VLGR1, and WHRN at the PMC is not required for the correct localization of these proteins. This suggests that SANS and HARM are more critical for the formation of PMC than USH2A, VLGR1, or WHRN. However, those three proteins may be significant in regulating OS protein transport through the CC.

Mutations in usherin were found to either cause USH2A or non-syndromic retinitis pigmentosa (nsRP) with preserved hearing[5]. In addition to the effect on the auditory organ, both diseases vary in the onset of visual impairment, with the onset in nsRP patients being 13

to 18 years delayed when compared to USH2A patients. In addition to this, the severity of the auditory phenotype of USH2A patients seems to be connected with the nature of the mutation. Truncating mutations displayed a more severe phenotype than non-truncating mutations[5,66]. In a recent study where comparisons of genotype and phenotype correlations were made of USH2A patients and were classified into three groups: those carrying two missense variants, those carrying one missense and one truncating variant, and those carrying two truncating variants[67]. Here, they provided evidence that vision loss in patients harboring two truncating variants has a much faster progression to low vision and legal blindness than those carrying two missense alleles or the combination of missense and truncating mutation. The medium age at legal blindness in homozygous missense patients was found to be at 54.5 years, while 52 years with missense and truncating mutation variants and 46 years with homozygous truncating mutation, thus demonstrating that truncating mutation such as c.2299delG result in a faster progression of the USH2A phenotype.

The work presented in this manuscript helps to refine our understanding of the functional role of usherin in the healthy and diseased retina. Our findings on the beneficial effects of expressing one WT usherin allele in the $Ush2a^{delG/+}$ support the use of gene augmentation or correction as the preferred strategy to rescue USH2A-associated visual defects. Photoreceptor function in the $Ush2a^{delG/+}$ was comparable to WT at all ages examined. Our data show that the presence of one WT copy of usherin is enough to promote normal retinal function, but also suggests that the WT usherin can help improve the localization/distribution of the mutant protein. This dual benefit supports gene augmentation as a therapeutic strategy for USH2A. Further investigations are needed to determine whether PMC localization of the truncated usherin helps restore vision or the presence of one WT usherin allele alone is sufficient. The reduced accumulation of the truncated protein in the ER of the $Ush2a^{delG/+}$ retina in comparison to $Ush2a^{delG/delG}$ suggests that the resulting cellular stress is less pronounced in the $Ush2a^{delG/+}$ retina, thus providing another possible explanation for a lack of retinal phenotype in $Ush2a^{delG/+}$ mice. It is also possible that the ability of the WT protein to overcome any abnormal functions of the truncated protein, or that reducing the amount of mutant protein by half (i.e., in the $Ush2a^{delG/+}$) is sufficient to improve the phenotype.

There are only a few examples of recessively inherited retinal diseases, which are connected to pathogenic gain-of-function mutations. The point mutations L945P and P858S in the guanylyl cyclase RetGC-1 expressing gene $GUCY2D$ causing autosomal recessive inherited Leber congenital amaurosis (LCA) were found to reduce the activity of wild type RetGC-1, thus representing an example of a dominant-negative mutation causing a recessively inherited disease[68]. For the mutant protein expressed in our $Ush2a^{delG/delG}$ model, we propose a combination of a loss-of-function and a mild gain-of-function effect. The comparison between the $Ush2a^{-/-}$ and the $Ush2a^{delG/delG}$ model shows that both models display comparable retinal phenotypes[21]. However, the $Ush2a^{delG/delG}$ model develops retinal phenotypes earlier than the knockout model. Thus, we conclude that the presence of truncated usherin actively deteriorates the health and functionality of the retina. This deterioration is partly caused by the prominent accumulation of the mutant usherin in the ER causing cellular stress. The mild dominant-negative effect of mutant usherin is further supported by our observation that $Ush2a^{delG/+}$ displayed a mild non-significant decrease in ERG responses as well as a portion of the mutant protein to accumulate in the ER. For the development of suitable gene therapy for USH2A future work using the knockin model will have to determine whether the reintroduction of the wildtype protein is sufficient to provide an efficient and long-lasting improvement in retina health and function, or whether the removal of the mutant protein is required.

In summary (Fig. 9), our study introduces a new model for USH2 carrying the most common USH2A mutation, c.2290delG (equivalent to human c.2299delG). The model exhibits a retinal phenotype associated with structural anomalies of the photoreceptor cilium and mislocalization of usherin interacting partners, VLGR1 and WHRN, as well as mislocalization of rod and cone opsins. The results obtained in this study prove that expression of the actual mutant protein is beneficial in reproducing USH2A retinal phenotype, and thus offers insight into strategies for designing therapeutic interventions. Assessment of the auditory phenotype in our model is necessary to fully validate that it is a representative of the human condition.

## Methods

### Reagents
A list of the antibodies used in this study is presented in Supplementary Table 1. All primers (Integrated DNA Technologies) used for qRT-PCR are listed in Supplementary Table 2. All reagents were purchased from Sigma-Aldrich unless stated otherwise.

### Generation of the $Ush2a^{delG/delG}$ KI mouse
The $Ush2a^{c.2290delG}$ KI mouse ($Ush2a^{delG/delG}$) was generated by inGenious Targeting Laboratory, Inc., (Ronkonkoma, NY, USA). The KI targeting vector incorporates the c.2290delG mouse mutation (equivalent to the human c.2299delG modification) and includes sequence coding for the extra 20 amino acids that occurs in the human mutant protein prior to the early stop codon. The murine KI sequence is followed by a 3x FLAG tag and a stop codon followed by an internal ribosome entry site (IRES) followed by GFP. The vector was targeted to the endogenous $Ush2a$ locus and knocked into exon 12 of the mouse $Ush2a$ gene located on chromosome 1. This vector was electroporated into embryonic stem cells followed by screening for the preferred allele. Positive clones were injected into blastocysts and implanted. Chimeric founders were bred to identify germline transmission of the targeted KI gene. Those found to retain the target gene in the germline were then bred to mice expressing the FLPeR transgene (Stock#003946, Jackson Labs, Bar Harbor, ME, USA) in order to remove the FRT-flanked Neo cassette that was used for selection of the ES cells. Genotyping using PCR confirmed that these mice are negative for the $rd8$ mutation. WT and KI animals also contain the L450 variant for RPE65[69]. Animals were backcrossed for four generations to obtain WT and $Ush2a^{delG/delG}$ littermates used in this study and all were on C57BL/6 J background. Since we did not observe any significant differences between the sexes, both male and female animals were used in all experiments and the data presented herein are reflective of both sexes. Euthanasia was performed via $CO_2$ asphyxiation followed by cervical dislocation. Animals were maintained in 12 L:12D cyclic light at ~30 lux unless otherwise specified in the experimental design. The mice were reared at an ambient temperature of 22 °C and a humidity of 50%. Animals were used under a protocol (protocol number: PROTO201800045) approved by the University of Houston Institutional Animal Care and Use Committee.

### Antibodies
Antibodies, their sources, and dilutions are listed in Supplementary Table 1.

### RNA preparation and analysis
Retinas at P1, P15, P30, and P360 were used to isolate total RNA using TRIzol reagent (Life Technologies, Grand Island, NY, USA) and treatment with RNase-free DNase I (Promega, Madison, WI, USA) according to manufacturer's recommendations. Superscript III reverse transcriptase (Life Technologies) along with oligo-dT primer plus random hexamer were used to perform reverse transcription[70]. Quantitative real-time PCR was done using 20 ng of cDNA in a real-time PCR detection system (C1000 Thermal Cycler, Bio-Rad Laboratories Inc., Hercules, CA, USA; software: CFX Manager, version 2.1.1022.0523)[71].

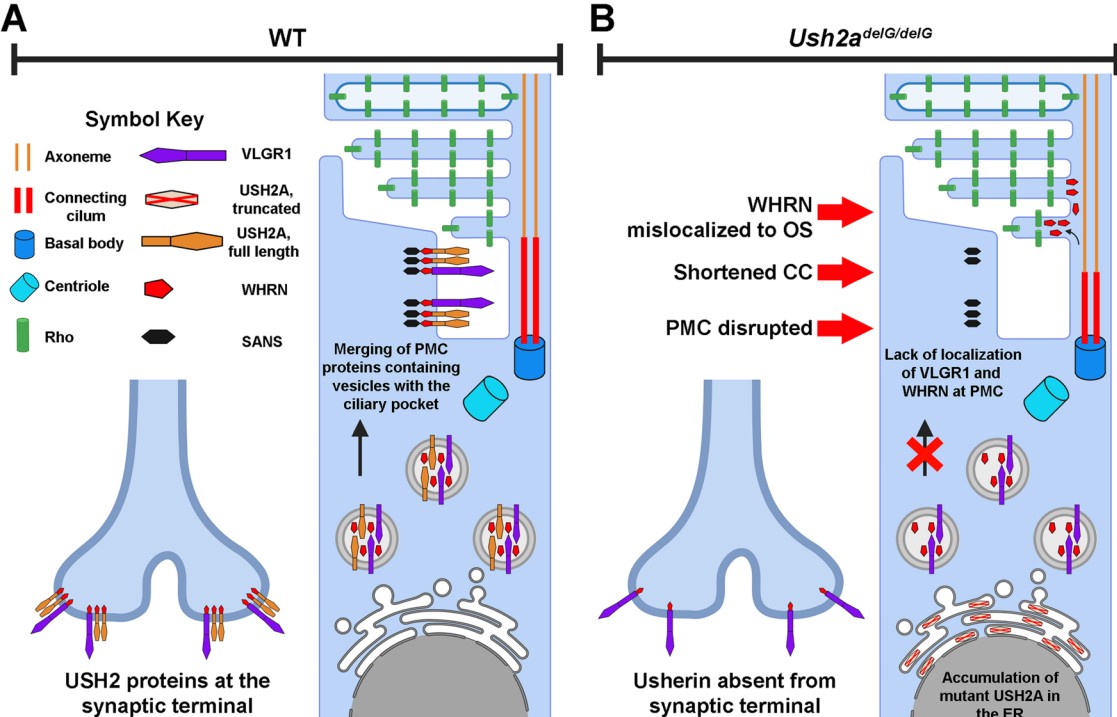

**Fig. 9 | Schematic summarizing the role of mutant usherin in impacting the structure of the periciliary membrane complex. A** In the WT photoreceptor cells, the PMC is intact, facilitating the efficient loading of cargo, including rhodopsin, into the CC. Rhodopsin is destined to the OS. **B** In the *Ush2a*$^{delG/delG}$ photoreceptor, the PMC is disrupted. WHRN and VLGR1 are trapped in the inner segment in vesicles, no longer able to localize at the ciliary pocket. WHRN is mislocalized towards the OS and inner segment. The disruption of the PMC decreases the loading efficacy of rhodopsin in the CC and thus impacts its transport toward the OS. Additionally, a huge portion of the mutant usherin is trapped in the ER, causing cellular stress. PMC periciliary membrane complex, OS outer segment, IS inner segment, Rho rhodopsin, ER endoplasmic reticulum, CC connecting cilium. Created with BioRender.com.

The qRT PCR was performed in triplicates with the ΔcT values calculated against the housekeeping gene hypoxanthine-guanine phosphoribosyl transferase (HPRT). Three independent samples were used for each time point and genotype. Primers to the N-terminal region of the *Ush2a* message present in both WT and KI transcripts, primers specific to the introduced mutation (i.e., KI only), and primers to the C-terminal region of the *Ush2a* message (i.e., endogenous only) were utilized. Sequencing primers are listed in Supplementary Table 2.

### Protein extraction and analysis

Retinas from WT and KI mice were collected and immediately frozen using liquid nitrogen and stored at −80 °C prior to processing. Extraction was done using 100 μl per two retinas of chilled (4 °C) RIPA buffer (1% NP-40, 0.1% SDS, 50 mM Tris, 150 mM NaCl, 0.5% sodium deoxycholate, 1 mM EDTA, pH 7.4) with 1x protease inhibitors (Roche Applied Sciences, Indianapolis, IN, USA). Samples were sonicated five times for 10 s on ice with 30 s incubation between each burst. Samples were then centrifuged for 5 min at 15,000×$g$ at 4 °C to remove any insoluble debris. The concentration of protein was evaluated using a colorimetric protein assay (Bradford reagent, Bio-Rad Laboratories Inc.). Samples were incubated at room temperature in 4x Laemmli buffer 400 mM Dithiothreitol (DTT) for 30 min on a shaker. For evaluation of intermolecular disulfide linkages, the extracts were either left untreated (non-reduced) or treated (reduced) with 400 mM DTT. This was followed by a 30 min incubation at room temperature in 4x Laemmli buffer ± DTT for 30 min before loading.

To determine glycosylation status, proteins were extracted in 1x PBS and 1% Triton X-100. One sample was left untreated (glycosylated) while the other sample was treated (deglycosylated) using a mixture of deglycosylation enzymes including PNGase F, Neuraminidase, β-*N*-Acetylglucosaminidase, β(1-4)Galactosidase, and *O*-Glycosidase according to manufacture recommendations (Protein deglycosylation

mix from New England Biolabs, Ipswich, MA, USA). SDS-PAGE was performed using an 8% gel and 1x running buffer (25 mM Tris-base, 200 mM Glycine, 0.1% SDS, pH 8.3). Immunoblotting was done at 25 V for 30 min using a Trans-Blot Turbo (Bio-Rad Laboratories Inc.) semi-dry blot system using 1x transfer buffer (25 mM Tris base, 200 mM Glycine, 0.1% SDS, 20% MeOH). Native PAGE was performed as in SDS-PAGE gels with the exclusion of SDS from the gel and running buffer, an 8% gel was used to resolve the mutant FLAG-tagged protein. Detection was performed with horse-radish peroxidase substrate (100 mM Tris-HCl pH 8.8, 1.25 mM luminol, 2 mM 4-iodophenylboronic acid (4IPBA), 5.3 mM H$_2$O$_2$)[72] on a ChemiDocTM MP imager (Bio-Rad Laboratories Inc.; Software: ImageLab 6.1, version 6.1.0 build 7). Primary antibodies were diluted in 5% non-fat dry milk as described in Supplementary Table 1. Secondary antibodies against mouse and rabbit conjugated to HRP diluted 1:15,000 in 5% non-fat dry milk were used.

### Immunofluorescence

For cryosections, eyes were enucleated following euthanasia, and fixed using 4% paraformaldehyde. After 2 h of fixation, the cornea and lens were removed. The eyecups were then further fixed for 2 h in 4% paraformaldehyde. Cryoprotection was achieved via a sequential sucrose gradient (10, 20, and 30%) before embedding in an M1 embedding medium (ThermoFisher, Waltham, MA, USA)[73,74]. Ten micrometer thick retinal cryosections were subjected to immunofluorescence (IF) analysis. The sections were washed for 10 min in 1x PBS followed by 10 min in H$_2$O. This was followed by 2 min in 1% NaBH$_4$ and 2 h blocking at room temperature (1x PBS, 5% BSA, 1% fish gelatin, 20% donkey serum, 1% Triton X-100). The sections were incubated overnight with the primary antibodies (diluted in blocking solution) at 4 °C. After washing three times in 1x PBS for 10 min each the sections were incubated for 2 h with the secondary antibodies diluted 1:1000 in

blocking solution at room temperature. Sections were then washed three times in 1x PBS and mounted in ProlongGold with DAPI (Invitrogen, Waltham, MA, USA). For cryosections showing the co-staining of HARM with acetylated tubulin and SANS with acetylated tubulin shown in Fig. 7A, B, as well as the ciliary length measurements in Fig. 5B, C, the eyes were processed as follows: eyes were enucleated following euthanasia, then whole eyes were dissected and cryoprotected as described above. Ten micrometer thick retinal cryosections were fixed for 10 min in acetone at −20 °C. After drying for 20 min, the sections were washed three times in 1x PBS for 10 min, blocked for 60 min in blocking solution (0.1% Triton X-100 and 5% FBS in 1x PBS) and incubated overnight with the primary antibodies at 4 °C. Primary antibodies were diluted in blocking solution. After incubation with the primary antibodies, the sections were washed three times with 1x PBS and incubated for 1.5 h with the secondary antibodies diluted 1:1000 in blocking solution at room temperature. The sections were then washed three times in 1x PBS and incubated for 30 min with DAPI (1:1000 diluted in blocking solution at room temperature). After two additional washes in 1x PBS the sections were mounted using Prolong Gold antifade mounting reagent (Thermo Fisher Scientific). The ciliary length measurements were performed in ImageJ (Version 1.53f51). For immunostaining displayed in Fig. 7C–F, eyes were collected after euthanasia and fixed for 4 h in 4% paraformaldehyde in 0.1 M sodium phosphate buffer pH 7.4 at 4 °C[75]. After fixation the eyes were transferred to 30% sucrose, embedded in OCT medium, and stored at −80 °C until they were completely frozen then four micrometer-thick sections were cut. After 10 min fixation with acetone at −20 °C, the sections were dried for 10 min. Rehydration was done in 1x PBS, followed by 30 min in blocking solution (0.1% Triton X-100 and 5% FBS in 1x PBS). The sections were then incubated overnight at 4 °C with the primary antibodies diluted in a blocking solution. This was followed by 3x washing in 1x PBS for 5 min. The sections were then incubated at room temperature for 1 h with the appropriate secondary antibody diluted in a blocking solution. Mounting was done using a Vectashield mounting medium (Vector Laboratories, Burlingame, CA, USA) combined with DAPI. For paraffin sections, the eyes were enucleated after euthanasia and incubated in Davidson fixative and paraffin-embedded (STP 120 Spin Tissue Processor, Thermo Scientific). The embedded eyes were sectioned in ten-micrometer sections (HM 355 S Automatic Microtome, Thermo Scientific) collected onto microscope slides, and stored at room temperature. For immuno-labeling the sections were deparaffinized with xylene, rehydrated in an ethanol gradient (100, 90, 80, 70, and 50%), and washed with water. For antigen retrieval, the slides were boiled for 30 min in Tris/EDTA buffer (10 mM Tris Base, 1 mM EDTA Solution, 0.05% Tween 20, pH 9.0) and cooled down at room temperature for 10 min. After antigen retrieval, the slides were washed twice with water and incubated for 3 min in 1% NaBH₄. This was followed by two additional washes in water and two washes in 1x PBS for 5 min each and by 30 min blocking (0.5% Triton X-100, 2.5% donkey serum, and 5% BSA in 1x PBS). The primary antibodies were diluted in a blocking solution. Sections were incubated with the primary antibodies overnight at 4 °C. After which the slides were washed three times in 1x PBS and incubated for 1.5 h with the secondary antibodies diluted 1:1000 in blocking solution. Following three washes in 1x PBS and 30 min incubation with DAPI (1:1000, at room temperature), sections were washed two more times in 1x PBS and mounted using Prolong Gold antifade mounting reagent (Thermo Fisher Scientific). All sections were imaged using a Zeiss LSM 800 (Carl Zeiss Microscopy GmbH, Jena, Germany; software: Zen 2.3, Version 2.3.64.0). The primary antibodies for all immunofluorescence experiments were diluted in the corresponding blocking solution as described in Supplementary Table 1. The following secondary antibodies were used: Alexa Fluor-488 donkey-anti-mouse, Alexa Fluor-555 donkey-anti-mouse, Alexa Fluor-647 donkey-anti-mouse, Alexa Fluor-488 donkey-anti-rabbit, Alexa Fluor-555 donkey-anti-rabbit and Alexa Fluor-647 goat-anti-

rabbit (all purchased from Invitrogen). Additionally, Alexa Fluor-488 goat-anti-chicken and Alexa Fluor-594 donkey-anti-rat (Life Technologies) were also used.

## Cone counts

Cone counts were performed using P500 retinas. Whole eyes were collected and fixed in 4% paraformaldehyde solution for 2 h. After 15 min of fixation, the eyes were punctured with a needle to allow fixative to enter the eye. One hour later, the eyes were dissected to remove the retina from the choroid and RPE. Retinas were fixed for an additional hour at 4 °C then were washed in Hank's balanced salt solution (HBSS) at pH 7.2 three times and then incubated in primary antibody (or PNA-488, 1:500 diluted in HBSS) for 1 h at room temperature. After 5x washing in HBSS, retinas were then cut with four small incisions, generating four petals, to allow the retina lay flat onto a slide and coverslipped. The photoreceptors were mounted toward the coverslip. Four 0.016 mm² areas (one in each petal) at a 300 μm distance from the optic nerve were imaged in a 60x window using the Olympus BX62 upright microscope equipped with a spinning disc confocal unit using a Hamamatsu C-4742 camera through UPlanSApo objectives (Olympus, Tokyo, Japan). Total cones were counted from three different WT and KI retinas (each from a different animal). The total number of cones from all four areas were averaged for each retina.

## Separation of soluble IPM, cytosolic, and membranes from retinal cells

Freshly dissected retinas from P30 WT or KI mice were incubated on ice in 1x PBS with 2x protease inhibitors for 20 min without agitation[76]. The insoluble interphotoreceptor cell matrix (IPM) was then separated from the soluble IPM (S-IPM) by centrifugation at 750×g for 5 min. The supernatant was considered to contain soluble components of the IPM. The pellet was washed three times in 1x PBS, resuspended in 0.1x PBS with 2x protease inhibitors and homogenized. The suspension was centrifuged at 50,000×g for 30 min in a Sorvall Discovery M150 ultracentrifuge (Thermo Scientific) equipped with a fixed angle rotor (Sorvall no. S55S-1009). The supernatant was considered to contain cytosolic components. The pellet was washed three times in 1x PBS followed by resuspension in 1x PBS with 1% Triton X-100. The pellet was then homogenized and sonicated with three 10 s bursts on ice with a 20 s incubation on ice between each burst. The resulting resuspension was incubated overnight at 4 °C on a rocking platform. The next day the resuspension was centrifuged for 10 min at 18,000×g and 4 °C. The supernatant contained the retinal membranes, organelles, cytoskeletal components, and insoluble extracellular matrix. This supernatant as well as the S-IPM and cytosolic fractions were analyzed via SDS-PAGE using a 12% gel and 1x running buffer (25 mM Tris-base, 200 mM Glycine, 0.1% SDS, pH 8.3). Immunoblotting was achieved via wet transfer using a Mini Trans-Blot Cell (Bio-Rad Laboratories Inc.) at 100 V for 90 min in 1x wet transfer buffer (25 mM Tris base, 200 mM Glycine, 0.01% SDS, 20% MeOH). Super Signal West Pico plus chemiluminescent substrate (Thermo Fisher) was used for detection on a ChemiDocTM MP imager (Bio-Rad Laboratories Inc.; Software: ImageLab 6.1, Version 6.1.0 build 7). Primary antibodies were diluted in 5% non-fat dry milk as described in Supplementary Table 1. Secondary antibodies against mouse and rabbit conjugated to HRP diluted 1:15,000 in 5% non-fat dry milk were used.

## Light and electron microscopy

Eyes were orientated via marking of the superior hemisphere along the vertical meridian at the limbus with a hot needle after euthanasia. A slit in the superior cornea was made and the eyes were then fixed for 2 h in 0.1 M sodium phosphate buffer (pH 7.4) containing 2.5% glutaraldehyde, 2% paraformaldehyde, and 0.025% CaCl₂ at 4 °C. The lens and superior cornea were removed followed by an additional fixation

overnight. This was followed by washing the fixed eyes in 0.1 M sodium cacodylate buffer (pH 7.4) containing 0.025% $CaCl_2$. Post fixation was achieved by incubation for 1 h in 1% osmium tetroxide one time in 0.1 M sodium cacodylate buffer and one time in distilled water. This was followed by dehydration in an ethanol gradient and propylene infiltration overnight in Spurr's resin. The eyes were embedded in resin-filled BEEM capsules and polymerized for 48 h at 70 °C. Using plastic-embedded semi-thin sections (1 μm) from P180, P360, and P500 whole eyes, the number of nuclei in the ONL in a 20x window was counted at 0.3, 0.9, 1.5, 2.1, 2.7, and 3.0 mm from the optic nerve head in both the inferior and superior directions using light microscopy. Images were taken with an Axioskop with an attached AxioCam ICc5 (Carl Zeiss Microscopy GmbH; software: Zen 2, Version 2.0.0.0). The number of nuclei was counted using ImageJ (version 1.53f51). For the measurement of OS length, an image was taken 300 μm inferior and another superior to the optic nerve from three independent retinas of three separate animals per genotype at P360. The length of the OS was measured using Zen 3.5 (version 3.5.093.00001, Carl Zeiss Microscopy GmbH). Thin sections (600–800 Å) were taken from whole eyes collected, fixed in 2% glutaraldehyde/4% paraformaldehyde, and processed for transmission electron microscopy (TEM)[77]. The thin section were placed on a copper 75/300 mesh grid. Staining was done using 2% uranyl acetate and Reynold's lead citrate. Images were taken with a JEOL 100CX electron microscope (JEOL Ltd., Tokyo, Japan) at an acceleration voltage of 60 kV.

### Connecting cilium width measurements
The width of the OS and the CC was measured from high-resolution TEM images from P360 WT and KI retinas. The measurements were done for 68 and 90 cilia of WT and KI mice, respectively. Measurements were performed using ImageJ (version 1.53f51).

### Optomotor response
Optomotor responses (OMR) were measured at P30, P180, and P500 using the Optometry apparatus (Cerebral Mechanics Inc., Alberta, Canada)[78]. Mice were positioned on a raised platform which was surrounded by four computer screens displaying vertical lines which rotated at a 12 d/s (degrees/second) drift speed, starting with a spatial frequency of 0.042 c/d (cycles/degree), and 100% contrast. The mice were observed via camera. Assessment of the tracking behavior was done blindly in response to exposure to stimuli at varying spatial frequencies. The measure of visual acuity was defined as the highest spatial frequency at which mice track the rotating cylinder. Since tracking is a temporal to nasal-specific reflex, counter-clockwise and clockwise tracking behavior test were used in the right and left eye, respectively.

### Electroretinography
Dark-adapted mice were anesthetized using 85 mg/kg ketamine and 14 mg/kg xylazine (Butler Schein Animal Health, Dublin, OH, USA)[71,79–81]. The eyes were then dilated, using 1% cyclogyl (Akorn, Lake Forest, IL, USA). Assessment of full-field ERG was done by using the UTAS system (LKC, Gaithersburg, MD, USA; software: EMwin, Version 9.8.0). Here, a platinum wire loop electrode was in contact with the cornea through a layer of Gonak (Akorn). The scotopic response was measured via a single strobe flash stimulus of 157 cd·s/m² presented to the dark-adapted mouse, followed by 5-min light adaptation with a background light intensity of 30 cd/m². Measurement of the photopic response was performed by averaging the responses from 25 flashes of white light at 157 cd·s/m². Intensity-response series of the scotopic ERG amplitude test in Fig. 2B was performed using similar conditions as indicated above. Postnatal 200 mice were dark-adapted overnight, anesthetized, pupils were dilated and full-field ERG was performed. The scotopic test was assessed with a single strobe flash stimulus of increasing light intensities, starting from 0.0045 cd s/m² all the way to 157 cd s/m², presented to the dark-adapted mouse. The single strobe flashes were presented sequentially followed by 1 min in the dark between the initial low-intensity flashes to 3–5 min for the high-intensity flashes.

### Study approval
All handling, maintenance, and experimental use of animals followed protocols approved by the University of Houston's Institutional Animal Care and Use Committees and were performed according to the NIH and the Association for Research in Vision and Ophthalmology (ARVO) guidelines.

### Statistics and reproducibility
All experiments, except where mentioned below, were reproduced in three independent experiments using independent samples. The fractionation experiments in Figs. 1J, K, 8J were performed once in triplicates on three independent mice for each genotype. Then reproduced twice using a fourth mouse of each genotype.

The immunohistochemical analysis in Fig. 1G, the co-staining of VLGR1 (Fig. 6G), and WHRN (Fig. 6H) with FLAG in the $Ush2a^{delG/delG}$ retina, the co-staining of HARM (Fig. 7A) and SANS (Fig. 7B) with acTub in both WT and $Ush2a^{delG/delG}$ retina, the co-labeling of VLGR1 and acTub in the $Ush2a^{delG/+}$ retina (Fig. 8C) and the co-labeling of WHRN and acTub in the $Ush2a^{delG/+}$ retina were all reproduced in four independent experiments.

The electron microscopy presented in Fig. 5G was reproduced in three independent experiments for the WT and in four independent experiments for $Ush2a^{delG/delG}$. The co-labeling of VLGR1 with acTub in the WT (Fig. 6C) and $Ush2a^{delG/delG}$ retina (Fig. 6D) were reproduced in six and seven independent experiments, respectively. The co-labeling of WHRN with acTub in the WT (Fig. 6E) and $Ush2a^{delG/delG}$ retina (Fig. 6F) were reproduced in five and four independent experiments, respectively.

**Statistical analyses**. Total RNA message levels measured using qRT-PCR were analyzed using a one-way ANOVA with Tukey's post hoc comparison, while the WT and mutant message levels were analyzed using a two-tailed unpaired $t$-test. OMR and ERG bar graphs were analyzed using a two-tailed unpaired $t$-test. Spider graphs showing the nuclei numbers counted from the ONL were analyzed using two-way ANOVA with Bonferroni's post hoc comparison. Cone counts were analyzed using two-way ANOVA with Bonferroni's post hoc comparison. The ciliary length was analyzed using a two-tailed unpaired $t$-test. Calculation of significance and plotting of all graphs was performed using Prism 7 (GraphPad Software Inc., Boston, MA, USA; version 7.05).

### Additional software
All figures except Fig. 9 were assembled using Photoshop 2020 (Adobe, San Jose, CA, USA; version 21.0.3). Figure 9 was assembled using the web tool BioRender (biorender.com). The qRT-data were sorted and calculated using Excel 2016 (Microsoft, Redmond, WA, USA; version 16.0.5378.1000) before the calculation of significance and assembling of the graph was performed using Prism 7 (GraphPad Software Inc.; software: version 7.05).

### Reporting summary
Further information on research design is available in the Nature Portfolio Reporting Summary linked to this article.

## Data availability
All data generated or analyzed during this study are included in this article and its supplementary information files. Source data are provided with this manuscript. Source data are provided with this paper.

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

## Acknowledgements

The authors thank Dr. Shannon Conley (University of Oklahoma Health Sciences Center) for her comments on the manuscript and Dr. Jun Yang at the University of Utah for sharing the usherin antibody specific to the C-terminus of the long isoform and the polyclonal chicken antibody

against VLGR1. We also thank Dr. Uwe Wolfrum at the Johannes Gutenberg University of Mainz for sharing several antibodies against different usher proteins listed in the manuscript. This project was supported by National Eye Institute (EY18656 and EY010609 to M.I.N. and M.R.A.-U. and EY034671 to M.I.N.) and departmental startup funds.

## Author contributions

L.T., M.L.M., M.R.A.-U., and M.I.N. designed and directed the research studies presented in this manuscript. L.T. and M.L.M. conducted experiments, analyzed data, and wrote the first draft of the manuscript. R.C. performed cone counts, M.S.M. performed ERG measurements and morphometry of *Ush2a*$^{delG/+}$, M.K. performed some of the immunohistochemistry shown in Figs. 1, 6, and 8, and D.C. performed some of the immunohistochemistry data presented in Fig. 7. M.R.A. and M.I.N. edited the manuscript to its final form.

## Competing interests

The authors declared no competing interests.
