## [Peer Review File · Nature Communications]

The usherin mutation c.2299delG leads to its mislocalization and disrupts interactions with whirlin and VLGR1REVIEWER COMMENTS

Reviewer #1 (Remarks to the Author):

This group developed and investigated a novel KI mouse by substituting a common human SNP variant (identified from affected human Usher syndrome type 2 populations) -- c.2299delG within USH2A. The authors developed this novel mouse model because the more commonly used Ush2a knock-out mouse model does not (they assert) recapitulate the structural and functional retinal anomalies of human Usher syndrome type 2.

Their results show that the novel KI model with a single SNP mutation resulted in a truncated/anomalous ush2a protein, leading to structural anomalies in the retinal ciliary complex, dislocation of photoreceptor opsins and ultimately, photoreceptor degradation associated with functional visual loss.

The retinal assessments obtained from the KI mice are diverse and detailed, including molecular (protein assay), structural (light and EM) and functional (OMR,ERG) measures. The assessments seem to have been carefully performed and the deviations in the KI model well documented. Having limited expertise in a number of techniques employed by the authors, I can offer little critique of Methods. The evidence for recapitulation of a retinal phenotype closely approximating the human Usher syndrome type 2 phenotype seems robust.

My concerns with the ms are more theoretical in nature.

Reviewer Comments

(1) Results from the current study indicate that truncation of the Ush2a protein is sufficient to disrupt the formation of the PMC, resulting in mislocation of opsins and photoreceptor degeneration. This is the major result, which the authors describe as novel. The novelty derives because the authors state that the more commonly used Ush2a KO model did not generate this phenotype (“.....Although[the Ush2a KO model] is useful for mimicking the hearing loss detected in patients and provided some insight into the function of usherin in the retina, IT DID NOT RECAPITULATE PHENOTYPE OBSERVED IN USH PATIENTS (Liu et al., 2007).”)

However, the authors elsewhere note that “...removing or altering any piece of this complex will ultimately lead to RP associated structural and/or functional defects.” And..... “... In the retina, usherin interacts with VLGR1, WHRN and SANS to form the periciliary membrane complex (PMC) essential for the loading of cargo at the photoreceptor cilium.... Knockout of either USH2A (Ush2a^{-/-}) or VLGR1 (Ush2c^{-/-}) led to the depletion of the other from the interactome in the periciliary region of the photoreceptor.”

Given these statements I am unclear on how and why would an ABSENCE of Ush2a NOT similarly disrupt the ciliary complex (and resulting photoreceptor opsin location and, thereby, function)?

I looked up the Liu et al. (2007) reference that the authors cite as evidence of a lack of retinal phenotype in the Ush2a KO. That paper specifically states that ..."In this study, we generated and analyzed a mouse model in which all known variants of usherin were ablated. Usherin-null mice developed a spectrum of retinal and hearing defects closely resembling those of USH2A human patients, which include progressive photoreceptor degeneration and moderate, nonprogressive hearing loss, especially at higher frequencies. Interestingly, before this study, no genetic mouse model of Usher syndrome has been shown to develop overt photoreceptor degeneration..”

In conclusion I am unclear on why the authors state (without explanation) that the visual phenotype for the more commonly used Ush2a KO model is invalid? At the very least, the authors should provide substantial additional discussion to explain WHY the Liu et al., 2007, results did not match the human

retinal phenotype in Usher syndrome type 2 as well as the current model. Indeed, unless there is a major flaw in the prior study (which the authors should then address), I don't think they can say the Ush2a KO model described in Liu et al. 2007 failed to recapitulate the retinal phenotype, whereas only the current model did so. This impacts that novelty of the current results.

(2) Why weren't auditory systems examined in this novel KI model? Do the authors intend further work to assess these systems (cochlear stereocilia, audiograms)? Confirmation of the full cross-modal (visual AND auditory) phenotype of Usher Syndrome type 2 would seem to be required in order to assert this novel KI mouse as a valid overall Usher Syndrome type 2 model.

Reviewer #2 (Remarks to the Author):

The manuscript by Tebbe et al reports an exciting new research model for the visual loss associated with Usher syndrome type 2A. Previous work in the field has reported few models with a robust retinal phenotype with which to study mechanisms and develop new therapies. The studies reported here, including molecular, biochemical, functional, and histological, are well designed and report a comprehensive characterization of the retinal and visual phenotype of a new clinical model for the most common mutation associated with Usher syndrome. Additionally, the manuscript is well written and easy to read.

I recommend a few minor concerns to improve the manuscript and broaden readership:

1. Significant statement: The authors claim "no treatment exists" for Usher, however rehabilitative treatments for the hearing loss (hearing aids, cochlear implants) and imbalance (physical therapy), are available. Please revise.

2. To improve the descriptive nature of the model nomenclature, change "Ush2aG/G" to "Ush2a^{delG}/^{delG}" (with the appropriate italics and superscript) throughout.

3. In a supplemental figure, include the differences in the mouse WT and KI mRNA sequences that highlight the del G and base pairs that encode the additional amino acids in the mutant protein.

4. One of the most exciting concepts discussed is whether the Usher disease mechanism is a loss-of-function, gain-of-function, dominant negative, or combination (also see #12). It would be good to note the absence of WT Usherin in KI retinas (Figure 1F left and middle panels).

5. Supplemental 1E and Figure 1J: to compare fractions that contain WT versus mutant Usherin, it would be easier if these were in the same figure.

6. Figure 2A: is the decrease from 13% to 15% statistically significant, thus supporting a "progressive decrease"?

7. Figure 2D, E, F: change y-axis to "Maximum Amplitude".

8. Figure 3B shows a loss of photoreceptors between P180 and P360 in KI retinas; but not between P360 and P500. For a progressively degenerative disease, what are possible explanations?

9. Figure 3C shows perinuclear accumulation of rhodopsin in KI retinas. Is Usherin known to bind/transport rhodopsin?

10. The terms "re-introduction" and "rescue" to describe the phenotype in heterozygous mice is confusing because it suggests treatment of a homozygous mutant, rather than having 1 normal copy being sufficient to prevent disease. Consider revising.

11. Discussion: include a brief summary of the Usher models that do have retinal/visual phenotypes. Including a brief mention of the USH1C and USH1F models that contain human Usher mutations and visual phenotypes would further support your hypothesis that mutant transcripts/proteins may contribute to disease.

12. Discussion: Recessive disorders are generally considered to result from a loss-of-function mechanism, whereas dominant disorders are considered to result from dominant-negative mechanisms. These are important concepts especially when considering therapeutic strategies. The authors suggest more than one mechanism may have direct and/or indirect effects in Usher, however, gene augmentation, which would not remove mutant transcripts or proteins, is also suggested as the preferred therapeutic strategy. Can you provide a more high level discussion on what is necessary versus sufficient to develop disease, as well as, needed to provide a therapeutic benefit? Are there examples of other recessive retinal diseases that are the result of a dominant negative mechanism?

13. In the second sentence of the third paragraph of the discussion, delete the repeated word "in".

14. The authors report that both male and female mice were used for each study. Sex as a biological factor is an important factor and now required to be addressed by funding agencies. Please include how many of each were used in each study.

Reviewer #3 (Remarks to the Author):

This work describes a novel mouse model to study Usher syndrome. The team generated a knock-in mouse expressing the most common mutation (c.2299delG) found in USH2A patients. Authors did a thorough job in analyzing the phenotype and presenting the results in informative figures. Data showing consequences for USH2A interacting proteins are appreciated and provide further information potentially explaining the retinal phenotype. The manuscript is well written, nicely presented and of interest to scientist working in the field. In contrast to the USH2A knockout mouse, the novel knockin mouse recapitulates aspects of the human retinal phenotype. As such, it would be a model for investigating disease mechanisms and testing therapeutic strategies. However, the phenotype is rather mild and manifests itself only in the aged mouse making investigations rather difficult.

Some critical points:

To compare the novel mouse model with the phenotype in patients, it would be interesting to learn about variations in the phenotype, age of onset and progression of disease in patients.

Please comment on the auditory phenotype – or the lack of it. USH2A knockouts are affected. It would thus be interesting (and relevant) to learn whether expression of the truncated protein induces a similar phenotype. Are other phenotypes to be expected? After all, USH2A seems expressed in various organs including kidney, testis, sperm and others. Do mice breed normally in a homozygous state?

Authors state that the mice carried the L450 variant in the Rpe65 gene. However, this variant is not normally found in C57BL/6 mice. Did the mice retain the L-variant after backcrossing the mice for 4 generations onto the C57BL/6 background?

Did authors observe any sex differences in the progression of the degeneration?

The knockin mouse expresses a truncated protein with additional 20 amino acids and a FLAG-tag. Although unlikely that the presence of the tag influences the phenotype, it might be prudent to mention this additional sequence in the discussion and that the protein differs from the endogenous protein found in patients.

Authors use a very bright flash to determine retinal function. This flash elicits a mixed rod/cone response in scotopic conditions. Even though authors provide photopic measurements (using the same intensity flash), it is not possible to judge the rod response with this experimental paradigm. Authors should present intensity series for both scotopic and photopic conditions. This would also allow to determine threshold intensities. Flicker ERGs might also be informative for the cone response.

Authors argue that they wanted to know whether the reduced scotopic ERG resulted from a reduction in the number of rod photoreceptors (p7). However, the scotopic recordings include a mixed response of cones and rods. Thus, counting the nuclei in cross sections (were cones included in the counts?) may not allow to conclusively answer the above question.

Can authors support / expand their findings with gene expression data? Of special interest are scRNA data of rods and cones. If those are not available, real-time PCR data on individual genes of interest (stress genes, genes involved in an inflammatory response, neuroprotective genes, etc) might allow a better description of the retinal reaction to the expression of the mutant protein in photoreceptors.

Ush2aG/+ mice. The text on p11 (last paragraph) may be a bit misleading since it may suggest a therapy. The wt allele was not re-introduced but was there from birth. Also, the presence of the wt allele did not really rescue the phenotype but rather did not lead to a phenotype. Unless authors can show a real rescue by re-introducing a wild type allele in the adult Ush2aG/G retina, I suggest rewording of the text.

Fig. 9: indicate shorter CC in the mutant photoreceptors?

Response to the reviewers' comments released on 7-28-2022:

We would like to thank the reviewers for their supportive comments and helpful suggestions to improve the current manuscript. We have taken all of these comments and suggestions into consideration when revising the manuscript.

We have tracked all changes in the text of the manuscript, so that the reviewers can view them easily. All the requested changes to the figures have also been made in response to suggestions by the reviewers, but they have not been tracked in order to reduce the size of the document.

Below is our response to each of the reviewer's comments:

Reviewer #1:

We thank this reviewer for taking the time to assess our manuscript.

(1) Results from the current study indicate that truncation of the Ush2a protein is sufficient to disrupt the formation of the PMC, resulting in mislocation of opsins and photoreceptor degeneration. This is the major result, which the authors describe as novel. The novelty derives because the authors state that the more commonly used Ush2a KO model did not generate this phenotype (".....Although[the Ush2a KO model] is useful for mimicking the hearing loss detected in patients and provided some insight into the function of usherin in the retina, IT DID NOT RECAPITULATE PHENOTYPE OBSERVED IN USH PATIENTS (Liu et al., 2007).")

However, the authors elsewhere note that "...removing or altering any piece of this complex will ultimately lead to RP associated structural and/or functional defects." And..... "... In the retina, usherin interacts with VLGR1, WHRN and SANS to form the periciliary membrane complex (PMC) essential for the loading of cargo at the photoreceptor cilium.... Knockout of either USH2A (Ush2a^{-/-}) or VLGR1 (Ush2c^{-/-}) led to the depletion of the other from the interactome in the periciliary region of the photoreceptor."

Given these statements I am unclear on how and why would an ABSENCE of Ush2a NOT similarly disrupt the ciliary complex (and resulting photoreceptor opsin location and, thereby, function)?

I looked up the Liu et al. (2007) reference that the authors cite as evidence of a lack of retinal phenotype in the Ush2a KO. That paper specifically states that ..."In this study, we generated and analyzed a mouse model in which all known variants of usherin were ablated. Usherin-null mice developed a spectrum of retinal and hearing defects closely resembling those of USH2A human patients, which include progressive photoreceptor degeneration and moderate, nonprogressive hearing loss, especially at higher frequencies. Interestingly, before this study, no genetic mouse model of Usher syndrome has been shown to develop overt photoreceptor degeneration.."

In conclusion I am unclear on why the authors state (without explanation) that the visual phenotype for the more commonly used Ush2a KO model is invalid? At the very least, the authors should provide substantial additional discussion to explain WHY the Liu et al., 2007, results did not match the human retinal phenotype in Usher syndrome type 2 as well as the current model. Indeed, unless there is a major flaw in the prior study (which the authors should then address), I don't think they can say the Ush2a KO model described in Liu et al. 2007 failed to recapitulate the retinal phenotype, whereas only the current model did so. This impacts that novelty of the current results.

Response: We have modified the wording of the 2nd and 4th paragraphs of the discussion to further discuss the differences between the two models and have changed some of the writing to avoid misinterpretation of data presented in the Ush2a^{-/-} paper and to stress the point of the late-onset retinal phenotype. A statement was added stressing that the main advantage of the Ush2a^{delG/delG} model over

the *Ush2a*^{-/-} model is that it shows an earlier onset of retinal defect, thus recapitulating the timeframe of disease progression in patients.

We thank the reviewer for taking the time to review the phenotype of the *Ush2a*^{-/-} presented by Liu et al. in their 2007 paper, and we are pleased to offer clarification on this issue. The *Ush2A*^{-/-} manuscript (Liu 2007) presented data only from a single very late timepoint (~20 months of age), and indicated that they saw no difference before 10 months. This is much later than retinal phenotypes seen in patients, which are pre- or post-pubescent in onset. In contrast, our findings show that in the c.2290delG model (which genetically mimics the human condition), the onset of the retinal phenotype is much earlier (~6 months of age, still young for a mouse), making the model more closely parallel to the phenotype seen in patients and confirming the validity of our model. In addition, the Liu 2007 paper only showed the basic phenotype (retinal thinning and ERG defect) at 20 months, and did not include any data to address potential cellular mechanisms, an area in which our model and our manuscript is able to provide insight. The formation of the truncated usherin protein in our knockin model and its mislocalization to the inner segment is a key finding. The absence of the truncated protein at the ciliary ridge mimics the case in the knockout model (i.e. it also has no usherin at the ciliary ridge), but the presence of the c.2290delG mutant protein in the cytosol is likely what accelerates the onset of retinal degeneration in our model, and suggests that the truncated protein may play a critical role in the time-course and mechanism of disease in patients. In follow up work using the *Ush2a*^{-/-} (Yang et al 2010) authors reported that the absence of usherin led to the absence of whirlin and VLGR1. However, in contrast, we find that when the truncated usherin mislocalizes from the ciliary ridge to the inner segment, whirlin and VLGR1 also mislocalize to the inner segment (rather than being absent). Combined this finding suggests that while usherin and its binding partners, whirlin and VLGR1, are needed for proper assembly of the periciliary ridge, and thus for the long-term health of photoreceptors. It is likely that the disease mechanisms in patients are likely more complex, reflecting both loss-of-function contributions from the lack of appropriately localized and assembled Usherin complexes at the ciliary ridge and gain-of-function contributions from the abnormal accumulation of the truncated usherin in the inner segment.

(2) Why weren't auditory systems examined in this novel KI model? Do the authors intend further work to assess these systems (cochlear stereocilia, audiograms)? Confirmation of the full cross-modal (visual AND auditory) phenotype of Usher Syndrome type 2 would seem to be required in order to assert this novel KI mouse as a valid overall Usher Syndrome type 2 model.

Response: Our knockin model shows a cochlear phenotype in addition to abnormalities listed in other organs, confirming some of the reported patient's symptoms. Using auditory functional testing, we generated data showing that this mouse model exhibits congenital hearing loss that persists throughout the life of the animal, similar to that seen in patients. We also show that the c.2290delG mutant allele leads to the expression of a truncated protein that is abnormally trapped inside the cell bodies of hair cells. In contrast to the truncated protein in the retina, the trafficking defect of the truncated usherin in the cochlea did not lead to mislocalization of its interacting partners, ADGRV1 (VLGR1) and whirlin. However, the defect did lead to stereocilia bundle disorganization, specifically in regions associated with the functional hearing loss. This disorganization in the stereocilia is also observed in early ages. Our cochlear findings in this model clarify the role of usherin in maintaining structural support, specifically in the longer IHCs, during the stereocilia developmental stages, which is crucial for the proper bundle organization and function of these hair cells.

The data we have accumulated addressing the cochlear phenotype and the underlying mechanism of the structural/functional defects of the stereocilia in the presence of c.2290delG is too much to be condensed into the current manuscript. The cochlea story will be published in a second manuscript that is currently in the final stages of preparation and will be submitted to Nature Communications for assessment. Following correspondence with the *Nature Communications* editorial team, it was agreed that the level of advance provided by description of the retinal phenotype included in the article, given the substantial amount of data included here, is sufficient. We have added points to the discussion

caveating that further analysis of the auditory phenotype will be required for complete validation of the mouse model.

Reviewer #2:

We appreciate the time taken by this reviewer to evaluate our manuscript and for the valuable suggestions listed below to improve on the readability and clarity of the presentation of our work. Below are our response to these comments:

1. *Significant statement: The authors claim "no treatment exists" for Usher, however rehabilitative treatments for the hearing loss (hearing aids, cochlear implants) and imbalance (physical therapy), are available. Please revise.*

Response: This reviewer is correct, that there are several treatment strategies for hearing loss/deafness. In fact, patients with Usher type 2 have benefited from hearing aids or cochlear implants to correct their hearing defects, but there are currently no approved genetic corrections/treatments available for the USH2A-retinopathy mainly due to many challenges associated with this disease. Namely, the broad spectrum of mutations, the large size of its cDNA hampering gene therapy development and limited knowledge on its pathogenicity. We have revised this statement to say "Usher syndrome is the most common form of inherited combined deafness and blindness, for which no genetic treatment currently exists to correct the underlying disease defects. While hearing aids and cochlea implants offer a viable therapy for the hearing loss caused by USH, no suitable therapeutic approach for the treatment of the retinopathy exists thus far (Toualbi, 2020). Development of a gene therapy approach is hampered by the large size of the cDNA of some USH protein, as well as by the broad spectrum of mutations."

2. *To improve the descriptive nature of the model nomenclature, change "Ush2aG/G" to "Ush2a^{delG}/delG" (with the appropriate italics and superscript) throughout.*

Response: Based on the request of this reviewer, we have changed the nomenclature to *Ush2a^{delG}/delG* throughout the manuscript and figures. We did not highlight these changes in the revision to avoid distraction.

3. *In a supplemental figure, include the differences in the mouse WT and KI mRNA sequences that highlight the del G and base pairs that encode the additional amino acids in the mutant protein.*

Response: This information was added to Supplemental Fig. S1.

4. *One of the most exciting concepts discussed is whether the Usher disease mechanism is a loss-of-function, gain-of-function, dominant negative, or combination (also see #12). It would be good to note the absence of WT Usherin in KI retinas (Figure 1F left and middle panels).*

Response: For the comment associate with the loss-of-function, gain of function or dominant negative, please see our response to point # 12 below. We have revised the text associated with Fig. 1F to stress this observation of the absence of WT usherin in the KI retinas as requested by this reviewer.

5. *Supplemental 1E and Figure 1J: to compare fractions that contain WT verses mutant Usherin, it would be easier if these were in the same figure.*

Response: We have combined these two items in Fig. 1 as requested by this reviewer.

6. *Figure 2A: is the decrease from 13% to 15% statistically significant, thus supporting a "progressive decrease"?*

Response: Yes, the reductions in optomotor response presented in Fig. 2A are significant at the two ages listed (P180 and P500).

7. *Figure 2D, E, F: change y-axis to "Maximum Amplitude".*

Response: Thanks, we have made this change in the revised manuscript.

8. *Figure 3B shows a loss of photoreceptors between P180 and P360 in KI retinas; but not between P360 and P500. For a progressively degenerative disease, what are possible explanations?*

Response: We noticed this as well and it likely reflects a very slow rate of retinal degeneration. While we do not have clear explanation to this at present, it has been well-established that functional declines (e.g. ERG and optomotor defects) do not directly parallel degeneration. For example, in many models, initial functional defects are much more severe than would be expected based on the cell loss (e.g. in the *Prph2^{+/-}* model of retinitis pigmentosa). In this case, it is possible that the progressive nature of the functional decline reflects ongoing defects in outer segment structure and/or length that proceed at a different pace than the cellular degeneration.

9. *Figure 3C shows perinuclear accumulation of rhodopsin in KI retinas. Is Usherin known to bind/transport rhodopsin?*

Response: Usherin among other proteins of the cilia are known to aid in trafficking outer segment cargos and rhodopsin is one of those. A short addition in the discussion stressing the importance of PMC members for rhodopsin transport was made. Rhodopsin mislocalization is a common feature observed in USH mouse and zebrafish models. However, the exact loading and transport mechanism of rhodopsin along the connecting cilium remains elusive. Given that usherin is restricted to the ciliary base and was never found to be localized along the connecting cilium, it seems unlikely, that it is mediating the actual transport from the base to the tip of the connecting cilium. It is more logical to mediate the cargo loading of rhodopsin at the base of the cilium. However, thus far there is no documented evidence of a direct interaction between Usherin and rhodopsin published.

10. *The terms "re-introduction" and "rescue" to describe the phenotype in heterozygous mice is confusing because it suggests treatment of a homozygous mutant, rather than having 1 normal copy being sufficient to prevent disease. Consider revising.*

Response: The wording was changed in the revised manuscript. The text now states the presence of one normal copy prevents the phenotypes observed in the homozygous *Ush2a^{delG/delG}* mice.

11. *Discussion: include a brief summary of the Usher models that do have retinal/visual phenotypes. Including a brief mention of the USH1C and USH1F models that contain human Usher mutations and visual phenotypes would further support your hypothesis that mutant transcripts/proteins may contribute to disease.*

Response: A paragraph providing a short overview of USH mouse models displaying a retinal phenotype was added to the discussion (4th paragraph of the discussion). A short comparison of published KO versus KI models (including the proposed KI models for USH1C and USH1F) was also included in the revision (2nd and 4th paragraph of the discussion).

12. *Discussion: Recessive disorders are generally considered to result from a loss-of-function mechanism, whereas dominant disorders are considered to result from dominant-negative mechanisms. These are important concepts especially when considering therapeutic strategies. The authors suggest more than one mechanism may have direct and/or indirect effects in Usher, however,*

gene augmentation, which would not remove mutant transcripts or proteins, is also suggested as the preferred therapeutic strategy. Can you provide a more high level discussion on what is necessary versus sufficient to develop disease, as well as, needed to provide a therapeutic benefit? Are there examples of other recessive retinal diseases that are the result of a dominant negative mechanism?

Response: We thank this reviewer for raising this point and we have added a paragraph in the discussion specifically addressing whether retinal phenotype in our knockin model suffers from loss-of-function and being recessive or a combination of loss-of-function and likely some side effect from the gain-of-function arising from the accumulation of the truncated protein. Data from our knockin homozygous versus heterozygous indicates that the recessive/loss-of-function effect is due to the absence of the truncated protein at the periciliary ridge as shown in the *Ush2a*^{-/-}. However, the more accelerated retinal degeneration and decrease in retinal function that we see in our model (and that mimics the patient case) likely reflects gain-of-function effects arising due to accumulation of the truncated usherin in the inner segment. This observation is supported by the very mild retinal phenotype seen in the heterozygous and that not all truncated usherin reach the connecting cilium, only small amount of it. This amount led to significant rescue though, a level that can be considered highly significant to patients.

13. In the second sentence of the third paragraph of the discussion, delete the repeated word "in".

Response: Thanks for bringing this error to our attention. It is corrected in the revised manuscript.

14. The authors report that both male and female mice were used for each study. Sex as a biological factor is an important factor and now required to be addressed by funding agencies. Please include how many of each were used in each study.

Response: In our initial studies of this model, we looked to see if there are any differences in retinal function and structure between *Ush2a* knockin males and females. These initial studies showed no differences and that is why in later studies we combined samples from both sexes. However, we observed reproductive systems differences and we are currently investigating the mechanism underlying the effect of c.2290delG mutation on these organs.

Reviewer #3:

We appreciate the glowing remarks by this reviewer on the quality of our manuscript.

Our response to the reviewer's comment on the mild phenotype seen in our model is to state that significant reduction we observe in scotopic and photopic ERG at P360 is comparable to that of patient's late onset retinitis pigmentosa. It is important to note that most currently available USH models do not mimic patient's onset of retinal phenotype and in some models retinal degeneration and functional defects only seen after high intensity light exposure. Retinal phenotype reported for our model is measured from animals exposed to cyclic light (12:12) with light cycle intensity around 70 lux.

Response to the critical points raised by reviewer 3

1. To compare the novel mouse model with the phenotype in patients, it would be interesting to learn about variations in the phenotype, age of onset and progression of disease in patients.

Response: A paragraph discussing the variability of disease progression and onset of the USH2A phenotype in patients was added to the discussion. In it, we stressed the fact that mutations in usherin can cause either syndromic USH2A or non-syndromic RP (nsRP, RP with no effect on the auditory function). Furthermore, it highlights that onset and progression of USH2A depend on the nature of the

underlying mutation, with truncating mutations resulting in an earlier onset and faster progression of both, retinal and auditory phenotypes.

2. Please comment on the auditory phenotype – or the lack of it. USH2A knockouts are affected. It would thus be interesting (and relevant) to learn whether expression of the truncated protein induces a similar phenotype. Are other phenotypes to be expected? After all, USH2A seems expressed in various organs including kidney, testis, sperm and others. Do mice breed normally in a homozygous state?

Response: Please see our response above to reviewer #1 addressing this issue.

3. Authors state that the mice carried the L450 variant in the Rpe65 gene. However, this variant is not normally found in C57BL/6 mice. Did the mice retain the L-variant after backcrossing the mice for 4 generations onto the C57BL/6 background?

Response: These mice were backcrossed for several generations onto the C57BL/6 genetic background and screened for the RPE65 L450 variant. Those that were heterozygous for this variant were crossed with each other's and *Ush2a*^{deG/delG} that are homozygous for the RPE65 L450 variant were selected for further crossing to ensure full C57BL/6 background that lacks RPE65 L450 variant.

4. Did authors observe any sex differences in the progression of the degeneration?

Response: We did not observe any phenotypic differences in the retina between males and females. However, we do observe some issues with the ovaries and sperm mortality in these animals that we are currently working on for future publication.

5. The knockin mouse expresses a truncated protein with additional 20 amino acids and a FLAG-tag. Although unlikely that the presence of the tag influences the phenotype, it might be prudent to mention this additional sequence in the discussion and that the protein differs from the endogenous protein found in patients.

Response: We do appreciate this comment and we have added this statement in the discussion.

6. Authors use a very bright flash to determine retinal function. This flash elicits a mixed rod/cone response in scotopic conditions. Even though authors provide photopic measurements (using the same intensity flash), it is not possible to judge the rod response with this experimental paradigm. Authors should present intensity series for both scotopic and photopic conditions. This would also allow to determine threshold intensities. Flicker ERGs might also be informative for the cone response.

Response: We have performed intensity series measurements for scotopic and photopic conditions and the data seems very interesting and states that the photoreceptors of these mice are sensitive to light. The ERG measurements increase significantly with successive exposure to increasing light intensities. We have expanded this observation at the cell, molecular and biochemical levels of experiments to address the mechanism of this observation. Collectively, we found that it is associated with the significant delay in the light-dependent translocation of transducin/arrestin that lead to the availability of some transducin (both of the alpha and beta subunits) in the outer segments of the light adapted retinas of these mice that allowed continues transduction. We also observed a significant delay in the recovery of the dark-adapted response after light adaptation that we were able to observe as early as 1 month old knockin animals. We are still investigating the underlying mechanism of the effect of usherin with the c.2299delG mutation on the rod and cone response to ERG at different light intensities. This work will be prepared for another manuscript.

7. Authors argue that they wanted to know whether the reduced scotopic ERG resulted from a reduction in the number of rod photoreceptors (p7). However, the scotopic recordings include a mixed response

of cones and rods. Thus, counting the nuclei in cross sections (were cones included in the counts?) may not allow to conclusively answer the above question.

Response: We have revised this statement to avoid any confusion and indicated that the reduction in the number of photoreceptors (rods and cones) could be responsible for the reduction in the associated function. The revised statement is “To determine whether the scotopic functional decline resulted from a reduction in the number of photoreceptors (rods and cones), morphometric analysis was performed on retinal sections taken through the optic nerve of P180, P360 and P500 *Ush2a*^{delG/delG} and WT eyes. We observed a statistically significant reduction (*p<0.05) in the number of photoreceptor nuclei starting at P360 in *Ush2a*^{delG/delG} mice, a difference that persists at P500 (Fig. 3A, B). The reduction in the number of photoreceptors was associated with perinuclear accumulation of rod (Fig. 3C) and cone (see below Fig. 4) opsins. Accumulation of rhodopsin became obvious as early as P60 *Ush2a*^{delG/delG} retinas and increased at P200 and P500 (Fig. 3C, examples highlighted by arrows).”

8. Can authors support / expand their findings with gene expression data? Of special interest are scRNA data of rods and cones. If those are not available, real-time PCR data on individual genes of interest (stress genes, genes involved in an inflammatory response, neuroprotective genes, etc) might allow a better description of the retinal reaction to the expression of the mutant protein in photoreceptors.

Response: These are very good suggestions and we are prepared to do these measurements as part of our next investigations. This manuscript is too large on its own and the current focus is on the mislocalization of the truncated usherin that led to the mislocalization of its interacting partners of VLGR1 and Whirlin.

9. Ush2aG/+ mice. The text on p11 (last paragraph) may be a bit misleading since it may suggest a therapy. The wt allele was not re-introduced but was there from birth. Also, the presence of the wt allele did not really rescue the phenotype but rather did not lead to a phenotype. Unless authors can show a real rescue by re-introducing a wild type allele in the adult Ush2aG/G retina, I suggest rewording of the text.

Response: We appreciate this thought and have revised this paragraph accordingly.

10. Fig. 9: indicate shorter CC in the mutant photoreceptors?

Response: We have added this information to Fig. 9.

REVIEWER COMMENTS

Reviewer #1 (Remarks to the Author):

The authors have addressed my concerns, and I appreciate them clarifying the timing distinctions in the phenotypes. This is very important information. I believe the manuscript as currently written is ready for publication.

Reviewer #2 (Remarks to the Author):

The manuscript by Tebbe et al describes a novel knock-in mouse model for the most common Usher mutation in the world, the USH2A c.2299delG mutation. The work includes molecular, structural, functional, and behavioral assessments that show a retinal phenotype with an earlier onset compared to the USH2A KO previously reported. Additionally, the data demonstrating in vivo structural deficits at an earlier age (shorter CC at P30) are new and important steps towards unraveling the mechanism of Usher retinal disease.

Overall, the revised manuscript is significantly improved with greater clarity and more comprehensive comparisons of previous work in the field.

I have a few minor comments:

1. Throughout the literature and in presentations at scientific meetings, differences in the definition and interpretation of a “retinal phenotype” in mouse models of Usher can be found. The additional discussion of the previous work with other models has greatly improved the manuscript that will be helpful to discern these differences to a broader audience of readers. Although not stated in the manuscript, the authors state in the response to the reviewers that the “retinal phenotypes in USH2A patients are pre- or post-pubescent in onset”. Additionally, when comparing the KI to the KO models, you state that “the onset of the retinal phenotype is much earlier (~6 months of age, ..) making the model more closely parallel to the phenotype seen in patients ..”. Do you consider the shorter CC at P30 to be a retinal phenotype? Typically, USH2A patients present (around the age of 15-18 years) with night blindness, which is detectable by ERG analyses. Using this logic (pre- or post-pubescent onset of night blindness), when would you expect the onset of a functional deficit in the mice? To better compare this model with other models and humans, and improve our understanding of the mechanisms of disease, it would be more helpful to discuss the time-course of deficits observed – structural first with shorter CC at P30, then behavioral with reduced optomotor responses at P180, molecular changes with the mis-localization of opsins at P200, functional and structural changes in ERGs and photoreceptor loss, respectively, at P360, etc.

2. At what age were the mice when the localization studies with interacting proteins VLGR1, WHRN, and other Usher proteins conducted, data presented in Figures 6 and 7? One might expect this to always be present, but for reproducibility, can you include the ages of mice tested?

3. Previously, I inquired whether “the decrease from 13% to 15% is statistically significant, thus supporting a “progressive decrease”, as shown in Figure 2A, to which you answered “Yes, the reductions are significant...” My question may not have been clear – I was not asking whether the responses between the KI and WT mice were significant at those ages, but rather whether the responses between the KI mice at P180 and P500 were different from each other? If these responses - the green bar at P180 compared to the green bar at P500 – are significantly different, can you indicate that in the figure and include the p-value? This would support the statement “This defect progressed slightly at P500 at which point we observed a 15.4% reduction when comparing Ush2adelG/delG to age-matched WT mice (Fig.2A).”

Reviewer #3 (Remarks to the Author):

It would help the reviewing process if authors included the position of all alterations made to the text in their response letter, especially since not all alterations were marked in red.

I do not understand the author's response to my 3rd point. If author selected Rpe65-L450 mice for backcrossing, why do they state: for further crossing to ensure full C57BL/6 background that LACKS RPE65 L450 variant? First, a 'full' BL/6 background would rather require the Rpe65 M450 variant – as far as I know. Second, by selecting the L450 variant, authors do not ensure a background that LACKS the L450 variant. Please clarify.

With the current ongoing discussion about gender medicine and the influence of sex on phenotypes in animal models, I urge authors to include a statement in the manuscript that they did not observe sex differences in the retinal phenotype – and to include a respective data set in supplemental data.

Authors respond to my inquiry about showing an intensity series for the determination of photoreceptor function. They state that they did the measurements and observed increased ERG responses with increasing light intensities, likely due to delayed translocation of transducin and arrestin. Given this statement, it seems that data presented in Fig 2 may mislead the reader into thinking that the progressive decline of function with age is the only difference to controls. Obviously, this is not correct, and this fact should be made clear in the manuscript. In addition, I feel that it is not best practice to show ERG data from only one light intensity. Therefore, I urge the authors to include the respective additional ERG data.

Response to the reviewers' comments

We would like to thank the reviewers for their supportive comments and helpful suggestions to improve the current manuscript. We have taken all of these comments and suggestions into consideration when revising the manuscript.

We have tracked all changes in the text of the manuscript, so that the reviewers can view them easily. All the requested changes to the figures have also been made in response to suggestions by the reviewers, but they have not been tracked in order to reduce the size of the document.

Below is our response to each of the reviewer's comments:

Reviewer #1:

The authors have addressed my concerns, and I appreciate them clarifying the timing distinctions in the phenotypes. This is very important information. I believe the manuscript as currently written is ready for publication.

Response: We thank this reviewer for seeing our revised version satisfactory for publication in Nature Communication.

Reviewer #2:

(1) Throughout the literature and in presentations at scientific meetings, differences in the definition and interpretation of a "retinal phenotype" in mouse models of Usher can be found. The additional discussion of the previous work with other models has greatly improved the manuscript that will be helpful to discern these differences to a broader audience of readers. Although not stated in the manuscript, the authors state in the response to the reviewers that the "retinal phenotypes in USH2A patients are pre- or post-pubescent in onset". Additionally, when comparing the KI to the KO models, you state that "the onset of the retinal phenotype is much earlier (~6 months of age, ..) making the model more closely parallel to the phenotype seen in patients ..". Do you consider the shorter CC at P30 to be a retinal phenotype? Typically, USH2A patients present (around the age of 15-18 years) with night blindness, which is detectable by ERG analyses. Using this logic (pre- or post-pubescent onset of night blindness), when would you expect the onset of a functional deficit in the mice? To better compare this model with other models and humans, and improve our understanding of the mechanisms of disease, it would be more helpful to discuss the time-course of deficits observed – structural first with shorter CC at P30, then behavioral with reduced optomotor responses at P180, molecular changes with the mis-localization of opsins at P200, functional and structural changes in ERGs and photoreceptor loss, respectively, at P360, etc.

Response: A timeline of the different retinal phenotypes observed in our KI model and the *Ush2a*^{-/-} model is added after paragraph 2 of the discussion (see line 405). The time course observed in the mouse models was compared to the time of onset for phenotypes in USH2 patients.

(2) At what age were the mice when the localization studies with interacting proteins VLGR1, WHRN, and other Usher proteins conducted, data presented in Figures 6 and 7? One might expect this to always be present, but for reproducibility, can you include the ages of mice tested?

Response: The localization studies were performed at P30. The information about the age was added to the result section (line 284) and the figure legends (Line 1122).

(3) Previously, I inquired whether “the decrease from 13% to 15% is statistically significant, thus supporting a “progressive decrease”, as shown in Figure 2A, to which you answered “Yes, the reductions are significant...” My question may not have been clear – I was not asking whether the responses between the KI and WT mice were significant at those ages, but rather whether the responses between the KI mice at P180 and P500 were different from each other? If these responses - the green bar at P180 compared to the green bar at P500 – are significantly different, can you indicate that in the figure and include the p-value? This would support the statement “This defect progressed slightly at P500 at which point we observed a 15.4% reduction when comparing *Ush2a*delG/delG to age-matched WT mice (Fig.2A).”

Response: The requested significance test between P180 and P500 OMR of the KI mice was performed. The difference was not significant. Thus, the wording in the results was changed from “This defect progressed slightly at P500...” to “This defect persisted at P500...” to prevent confusion.

Reviewer #3:

(1) It would help the reviewing process if authors included the position of all alterations made to the text in their response letter, especially since not all alterations were marked in red.

Response: We apologize for not including the changes we made in the letter. We have included the changes for the current revision in the response letter and we have also included the lines in the manuscript where the changes are located. To make it easier to identify the earlier changes on the manuscript are underlined and the changes for the current revision are bolded. To simplify, we did not mark minor editorial and grammatical changes that were recommended by the reviewers.

Based on the request of this reviewer, we included below all changes we made on the first revisions:

Line 49-Significance Statement: We introduced “While hearing aids and cochlea implants offer a viable therapy for the hearing loss caused by USH2, suitable therapeutic approach for the treatment of the retinopathy is unattainable. Development of a gene therapy approach is hampered by the large sized cDNA of some USH proteins, as well as the broad spectrum of mutations.”

Line 427-Discussion: We introduced “The formation of the truncated usherin protein in our KI model and its mislocalization to the inner segment is a key finding. Furthermore, the absence of the truncated protein at the ciliary ridge in our c.2290delG model mimics observations made in the *Ush2a*^{-/-} model (i.e. it also has no usherin at the ciliary ridge), but the presence of the c.2290delG mutant usherin in the cytosol is likely what accelerates the onset of retinal degeneration in our model, and suggests that the truncated protein may play a critical role in the time-course and mechanism of disease in patients. In a follow up work using the *Ush2a*⁻¹², authors reported that the absence of usherin led to the absence of WHRN and VLGR1 from the PMC. In contrast, we find that when truncated usherin is mislocalized from the ciliary ridge to the inner segment, WHRN and VLGR1 also mislocalize to the inner segment (rather than being absent). Combined these findings suggest that usherin and its partners, whirlin and VLGR1, are needed for proper assembly of the periciliary ridge. Although this assembly is crucial for the long-term health of photoreceptors, disease mechanisms in patients are likely more complex, reflecting both loss-of-function contributions from the lack of appropriately localized and assembled usherin

complexes at the ciliary ridge and gain-of-function contributions from the abnormal accumulation of the truncated usherin in the inner segment. The earlier onset of the retinal phenotype observed in the KI model in comparison to the *Ush2a*^{-/-} was an expected outcome since many USH2A mutations are proposed to generate truncated proteins.”

Line 458-Discussion: We introduced “. In addition to the 20 amino acids, our model also contains a 3x FLAG tag, which represents a difference between the protein expressed in our model, and the mutant protein expressed in patients. However, given that thus far no evidence of misfolding caused by this commonly used tag has been described, it seem rather unlikely that the ER accumulation observed in the KI mice is caused by the presence of the 3x FLAG tag.

The importance of having a truncated or mutant protein expressed in order to recapitulate the retinal phenotype is supported by the fact that most knockout models of USH proteins fail to display effects in the retina (reviewed in⁴¹). Apart from the above described *Ush2a*^{-/-} model, only a USH2D model in which the long isoform of whirlin is specifically knocked out displayed a robust retinal phenotype¹². However, just like in the *Ush2a*^{-/-} model this retinal phenotype only manifests in old mice (between 28 and 33 months of age). In line with our KI model and the *Ush2a*^{-/-} model, the PMC is also disrupted in this USH2D model. For *Ush3a*^{-/-} mice (clarin-1) conflicting results were obtained, with a recent study proving a significant decrease in ERG responses observed as early as P90⁴⁸, while an older study could not find any retinal phenotype⁴⁹. However, retinal degeneration was absent in both studies. Studies utilizing mouse models in which the mutant protein is expressed were more successful in reproducing the retinal phenotype. A study analyzing mice with different mutations in the Myo7a (USH1B) allele found a decreased ERG response in several of these mutant models starting as early as P70⁵⁰. An USH1C knockin model expressing mutant HARM (*Ush1c*^{c.216>A}) successfully reproduced the retinal phenotype observed in USH1C patients⁵¹. Here, a significant reduction in ERG responses as well as retinal degeneration could be observed (onset at P30 and P180, respectively). A second knockin with a prominent retinal phenotype was the USH1F model (*Ush1f*^{R250X}) expressing a truncated version of PCDH15⁴⁰. This model displayed an early onset reduction in ERG responses at P30 and retinal degeneration between 12-14 months of age.”

Line 536-Discussion: We introduced “The importance of the PMC and the correct localization of USH proteins in it for rhodopsin transport is further supported by observations made in the whirler mouse (USH2D model)²⁰ and a mouse model carrying a mutation in MYO7A (USH1B model)⁶³. Both proteins were found to interact with usherin at the ciliary base^{12,17} and the disruption of the PMC observed in these mice coincided with rhodopsin mislocalization.”

Line 552-Discussion: We introduced “Mutations in usherin were found to either cause USH2A or non-syndromic retinitis pigmentosa (nsRP) with preserved hearing⁵. In addition to the effect on the auditory organ, both diseases vary in the onset of visual impairment, with the onset in nsRP patients being 13 to 18 years delayed when compared to USH2A patients. In addition to this, the severity of the auditory phenotype of USH2A patients seems to be connected with the nature of the mutation. Truncating mutations displayed a more severe phenotype than non-truncating mutations^{5,66}. In a recent study where comparisons of genotype and phenotype correlations were made of USH2A patients and were classified into three groups: those carrying two missense variants, those carrying one missense and one truncating variant, and those carrying two truncating variants⁶⁷. Here, they provided evidence to show that risk of visual decline in patients harboring two truncating variants have a much faster progression to low vision and legal blindness than those carrying two missense alleles or the combination of missense and truncating mutation. The medium age at legal blindness in homozygous missense patients was found to be at 54.5 years, while 52 years with missense and truncating mutation variants and 46 years with homozygous truncating mutation, thus demonstrating that truncating mutation such as

c.2299delG result in a faster progression of the USH2A phenotype.”

Line 584-Discussion: We introduced “There are only few examples of recessively inherited retinal diseases, which are connected to pathogenic gain-of-function mutations. The point mutations L945P and P858S in the guanylyl cyclase RetGC-1 expressing gene *GUCY2D* causing autosomal recessive inherited Leber congenital amaurosis (LCA) were found to reduce the activity of wild type RetGC-1, thus representing an example of a dominant-negative mutation causing a recessively inherited disease⁶⁸. For the mutant protein expressed in our *Ush2a*^{delG/delG} model we propose a combination of a loss-of-function and a mild gain-of-function effect. The comparison between the *Ush2a*^{-/-} and the *Ush2a*^{delG/delG} model shows that both models display comparable retinal phenotypes²¹. However, the *Ush2a*^{delG/delG} model develops retinal phenotypes earlier than the knockout model. Thus, we conclude that the presence of the truncated usherin actively deteriorates the health and functionality of the retina. This deterioration is partly caused by the prominent accumulation of the mutant usherin in the ER causing cellular stress. The mild dominant-negative effect of mutant usherin is further supported by our observation that *Ush2a*^{delG/+} displayed a mild non-significant decrease in ERG responses as well as a portion of the mutant protein to accumulate in the ER. For the development of a suitable gene therapy for USH2A future work using the knockin model will have to determine whether the reintroduction of the wildtype protein is sufficient to provide an efficient and long-lasting improvement in retina health and function, or whether the removal of the mutant protein is required.”

Line 608-Discussion: We introduced “Assessment of the auditory phenotype in our model is necessary to fully validate that it is a representative of the human condition. To that end, we have fully investigated the cochlear changes in this model and found that mice exhibited congenital hearing loss associated with disorganized stereocilia.”

CURRENT REVISION (bolded in manuscript)

Line 284-Results: We introduced “To analyze the localization of these usherin interactors, IF was performed on P30 WT and *Ush2a*^{delG/delG} retinal sections.”

Line 405-Discussion: We introduced “The early onset retinal changes we observed in our KI model include shortening of the CC and mislocalization of the core PMC components of mutant usherin, VLGR1 and WHRN. These changes manifest as early as P30. While the ciliary structure was not analyzed in *Ush2a*^{-/-} mice, a disruption of the PMC in the retina could also be shown in the *Ush2a*^{-/-} retina, however the earliest time point at which this disruption was seen was not stated¹². An additional difference to our model is that in the *Ush2a*^{-/-} retina, both WHRN and VLGR1 are depleted rather than mislocalized. A second retinal phenotype displayed by our KI model was the mislocalization of rhodopsin and cone opsins starting at P60 and P200, respectively. Although rhodopsin mislocalization was not reported for the *Ush2a*^{-/-} model, cone opsins were mislocalized as early as P80^{21,42}. While data on CC structure and localization of opsins and PMC components are not available for patients, functional decline in the retina in USH2A patients manifest pre- or post-pubescence, usually in the second decade of life^{3-5,43}. The first sign of a decline in visual capabilities in the KI mice was observed at P180 (comparable to second decade of life in patients) with a significant decrease in OMR. ERG responses gradually decreased as the animals got older and reached statistical significance at P360. This is different from USH2 patients, in whom decreased ERG responses are observed at a very young age³. However, when compared to the *Ush2a*^{-/-} model which displays decreased ERG responses only at 20 month of age, our KI model performs better in reproducing the retinal phenotype of patients²¹. In line with this, our KI model showed a significant thinning of the ONL as early as P360, while ONL thickness remained unaffected in the *Ush2a*^{-/-} model as late as 20 month of age.

”

Line 1122-Legend of Figure 6: We introduced “P30 WT and *Ush2a*^{delG/delG} retinas were used.”

(2) I do not understand the author’s response to my 3rd point. If author selected Rpe65-L450 mice for backcrossing, why do they state: for further crossing to ensure full C57BL/6 background that LACKS RPE65 L450 variant? First, a ‘full’ BL/6 background would rather require the Rpe65 M450 variant – as far as I know. Second, by selecting the L450 variant, authors do not ensure a background that LACKS the L450 variant. Please clarify.

Response: We apologize for not clarifying this issue in the first time as we have been using the same strategy for all our knockin models that we have previously published. The C57BL/6 genetic background has nothing directly to do with RPE65 variants. All C57BL/6 strains available by vendors have the RPE65 M450 variant. In our lab, we have generated, through backcrossing, inbred wild-type C57BL/6 mice that lack the *rd8* mutation and carry the RPE65-L450 variant. The KI mice were initially on a C57BL/6 with the M450 variant. Therefore, the KI mice were backcrossed into our C57BL/6 mice to generate the final mice without the *rd8* and the methionine variant of RPE65.

(3) With the current ongoing discussion about gender medicine and the influence of sex on phenotypes in animal models, I urge authors to include a statement in the manuscript that they did not observe sex differences in the retinal phenotype – and to include a respective data set in supplemental data.

Response: **Line 632-Method Section:** We introduced “Since we did not observe any significant differences between the sexes, both male and female animals were used in all experiments and data presented herein are reflective of both sexes.”

(4) Authors respond to my inquiry about showing an intensity series for the determination of photoreceptor function. They state that they did the measurements and observed increased ERG responses with increasing light intensities, likely due to delayed translocation of transducin and arrestin. Given this statement, it seems that data presented in Fig 2 may mislead the reader into thinking that the progressive decline of function with age is the only difference to controls. Obviously, this is not correct, and this fact should be made clear in the manuscript. In addition, I feel that it is not best practice to show ERG data from only one light intensity. Therefore, I urge the authors to include the respective additional ERG data.

Response: We observed increased ERG responses when animals were repeatedly flashed with high intensity light. This matter is currently under investigation and we hope to describe this in the next manuscript detailing the functional and biochemical analyses addressing this increase in response. For this revision, we included scotopic ERG responses with increasing light intensities performed on KI and WT mice before the onset of degeneration (P200). Amplitude of dark-adapted a- and b-waves from 10 animals were plotted as a function of flash luminance from KI and WT and the results showed that the KI mice have the same ERG amplitudes and kinetics as those recorded from age matched WT mice. This information is included in Fig. 2B. The text in the result section for this figure was modified to state this observation (please see line 202 to 207, highlighted in yellow) and the method section was also modified and highlighted in yellow (see line 793 to 799).

REVIEWERS' COMMENTS

Reviewer #2 (Remarks to the Author):

The authors have sufficiently addressed my concerns. The additional work to clarify the experimental designs and interpretations will greatly benefit a broad audience of readers. I recommend publication in it's current version.

Reviewer #3 (Remarks to the Author):

I thank the reviewers for their clarifying statements and the inclusion of the ERG intensity series at baseline (before degeneration). I am very much looking forward to the results about the increased ERG responses in a future paper. It is clear that not all questions can be answered in one single manuscript.

I have no further concerns or questions